# Tricalbin proteins regulate plasma membrane phospholipid homeostasis

Ffion B Thomas[1], Deike J Omnus[1], Jakob M Bader[1], Gary HC Chung[1], Nozomu Kono[2], Christopher J Stefan[1]

The evolutionarily conserved extended synaptotagmin (E-Syt) proteins are calcium-activated lipid transfer proteins that function at contacts between the ER and plasma membrane (ER-PM contacts). However, roles of the E-Syt family members in PM lipid organisation remain incomplete. Among the E-Syt family, the yeast tricalbin (Tcb) proteins are essential for PM integrity upon heat stress, but it is not known how they contribute to PM maintenance. Using quantitative lipidomics and microscopy, we find that the Tcb proteins regulate phosphatidylserine homeostasis at the PM. Moreover, upon heat-induced membrane stress, Tcb3 co-localises with the PM protein Sfk1 that is implicated in PM phospholipid asymmetry and integrity. The Tcb proteins also control the PM targeting of the known phosphatidylserine effector Pkc1 upon heat-induced stress. Phosphatidylserine has evolutionarily conserved roles in PM organisation, integrity, and repair. We propose that phospholipid regulation is an ancient essential function of E-Syt family members required for PM integrity.

## Introduction

Maintaining the mechano-chemical properties of the plasma membrane (PM) is essential to vital processes including selective ion and nutrient transport, as well as size and shape control in all living cells. Accordingly, the PM has a distinctive lipid composition in eukaryotic cells, including high sterol and sphingolipid content as well as an enrichment of phosphatidylserine (PS) in its cytosolic leaflet that endows the PM with its unique identity, biophysical properties, organisation, and integrity (Schneiter et al, 1999; van Meer et al, 2008; Yeung et al, 2008; Lingwood & Simons, 2010; Bigay & Antonny, 2012; Holthuis & Menon, 2014). PM lipid composition is achieved and maintained, as needed, through the selective delivery of lipids from the ER where they are synthesized to the PM by vesicular and non-vesicular transport pathways. Vesicular lipid trafficking occurs via the secretory pathway alongside PM-bound proteins (Klemm et al, 2009; Fairn et al, 2011). It is also clear that lipid transfer proteins mediate non-vesicular lipid exchange between the ER and PM in the control of PM lipid composition and homeostasis (Kaplan & Simoni, 1985; Urbani & Simoni, 1990; Vance et al, 1991; Wong et al, 2019). Membrane contact sites between the ER and the PM, termed ER–PM contacts, are proposed to serve as integral sites for the coordinated regulation of lipid metabolism and transport (Pichler et al, 2001; Chang et al, 2017; Balla et al, 2020; Nishimura & Stefan, 2020; Reinisch & Prinz, 2021). However, a strict requirement for ER–PM contacts in non-vesicular transport of lipids from the ER to PM has been recently questioned (Quon et al, 2018; Wang et al, 2020), necessitating further evaluation of the vital roles of ER–PM contacts, as well as the proteins proposed to form and function at these important cellular structures.

Whilst roles of several lipid transfer proteins identified at ER–PM contacts are established, functions of the extended synaptotagmin (E-Syt) protein family members are incompletely understood and even controversial. The E-Syt proteins, as well as their budding yeast orthologs, named tricalbins, are anchored in the ER membrane via a N-terminal hairpin anchor (Giordano et al, 2013) and interact with the PM in a phosphoinositide lipid- and $Ca^{2+}$-dependent manner via their multiple C-terminal cytoplasmic C2 domains (Chang et al, 2013; Giordano et al, 2013; Idevall-Hagren et al, 2015; Saheki et al, 2016; Bian et al, 2018). They also feature a central cytosolic synaptotagmin-like, mitochondrial (SMP) domain that dimerizes and contains a deep hydrophobic groove previously shown to bind and transport lipids in vitro (Schauder et al, 2014; Saheki et al, 2016; Yu et al, 2016; Bian et al, 2018; Bian & De Camilli, 2019). While cellular roles of E-Syt proteins as ER–PM tethers are described (Giordano et al, 2013; Fernández-Busnadiego et al, 2015), roles of the E-Syts in membrane lipid dynamics in vivo are enigmatic. One study proposed a role of the E-Syts in the transfer of diacylglycerol from the PM to the ER during the phosphoinositide cycle, but loss of the E-Syt1/2/3 proteins had no significant effect on phosphoinositide lipid synthesis or the homeostasis of other

[1]Medical Research Council Laboratory for Molecular Cell Biology, University College London, London, UK    [2]Department of Health Chemistry, Graduate School of Pharmaceutical Sciences, The University of Tokyo, Tokyo, Japan

Correspondence: c.stefan@ucl.ac.uk
Deike J Omnus's present address is Science for Life Laboratory, Department of Molecular Biosciences, The Wenner-Gren Institute, Stockholm University, Stockholm, Sweden.
Jakob M Bader's present address is Department of Proteomics and Signal Transduction, Max Planck Institute of Biochemistry, Martinsried, Germany.

phospholipids at the PM (Saheki et al, 2016). An earlier study found that depletion of the E-Syt1/2 proteins impaired the re-synthesis of phosphatidylinositol (4,5)-bisphosphate, commonly termed PI(4,5)P$_2$, during the phosphoinositide cycle and suggested a role of E-Syt–mediated ER–PM contacts in the transfer of phosphatidylinositol from the ER to the PM (Chang et al, 2013). Yet another study has even suggested a role for E-Syt2 in PI(4,5)P$_2$ turnover (Dickson et al, 2016). While the findings in these studies are not necessarily mutually exclusive or contradictory, they highlight unresolved issues regarding the roles of the E-Syts in PM lipid homeostasis.

The budding yeast E-Syt orthologs, the tricalbins (Tcb1/2/3), have also been shown to play a role in ER–PM contact formation (Manford et al, 2012; Toulmay & Prinz, 2012; Collado et al, 2019; Hoffmann et al, 2019). Two recent studies used advanced cryo-electron tomography to reveal peaks of extreme ER membrane curvature at Tcb-mediated ER–PM contacts (Collado et al, 2019; Hoffmann et al, 2019). In particular, one study also found that Tcb-dependent ER–PM contacts are induced upon PM stress conditions and required to maintain PM integrity upon stress conditions (Collado et al, 2019). Moreover, computational modelling suggested that the region of extreme ER membrane curvature formed at Tcb-mediated ER–PM contacts may facilitate lipid transfer from the ER to the PM (Collado et al, 2019). However, potential functions of the Tcb proteins at ER–PM contacts remain to be experimentally tested. Consequently, insight into the roles of the Tcb proteins in PM lipid homeostasis and integrity is lacking.

In this study, we use quantitative lipidomics and microscopy approaches to elucidate roles of ER–PM contacts and the Tcb proteins in the control of PM lipid composition. The results suggest a role of ER–PM contacts in the delivery of mono-unsaturated phosphatidylserine (PS) and phosphatidylethanolamine (PE) species, but not phosphatidylinositol (PI), from the ER to the PM. The data also show that the Tcb proteins regulate phospholipid homeostasis at the PM upon stress conditions. Furthermore, we find that Tcb3 co-localises with the PM protein Sfk1, an ortholog of the mammalian TMEM150/FRAG1/DRAM proteins, that is implicated in stress-induced PI(4,5)P$_2$ synthesis and phospholipid asymmetry at the PM (Audhya & Emr, 2002; Chung et al, 2015; Mioka et al, 2018; Kishimoto et al, 2021). Finally, we find that the Tcb proteins promote the recruitment of Pkc1 to the mother cell cortex upon heat stress conditions. Altogether, our findings indicate that the Tcb proteins function as inducible ER–PM tethers necessary for PM phospholipid homeostasis upon stress conditions, providing new insight into Tcb protein function at ER–PM contact sites.

# Results

### The tricalbins are not required for delivery of phosphatidylinositol to the PM

Several proteins form and function at ER–PM contacts in yeast, including Scs2/22 (VAP orthologs), Ist2 (ANO8/TMEM16 ortholog), Lam1-4 (GRAMD1/Aster orthologs), and the tricalbin (Tcb) proteins (Loewen et al, 2007; Manford et al, 2012; Gatta et al, 2015) (Fig 1A). A primary aim of this study is to elucidate functions of the Tcb

proteins at ER–PM contacts that remain incompletely understood. Under normal growth conditions, loss of Tcb1, Tcb2, and Tcb3 (in tcb1/2/3Δ cells) does not result in obvious effects on ER–PM tethering (Manford et al, 2012) or phospholipid and calcium (Ca$^{2+}$) homeostasis (Figs 1 and 2, explained in further detail below). However, roles of the Tcb proteins in ER–PM tethering become apparent upon loss of additional proteins that form ER-PM contacts, including Scs2/22 and Ist2 (Fig 1A) (Manford et al, 2012; Collado et al, 2019; Hoffmann et al, 2019). Specifically, Tcb-mediated ER–PM contacts are induced upon loss of the Scs2/22 and Ist2 proteins (in scs2/22Δ ist2Δ triple mutant cells) (Collado et al, 2019), and loss of Tcb1/2/3 in combination with loss of Scs2/22 and Ist2 results in additive defects in ER–PM tethering (Manford et al, 2012). However, specific roles of the Tcb proteins in PM lipid homeostasis have not been thoroughly examined. We therefore monitored the consequential effects of loss of the Tcb proteins in scs2/22Δ ist2Δ triple mutant cells on PM lipid homeostasis by examining scs2/22Δ ist2Δ cells versus scs2/22Δ ist2Δ tcb1/2/3Δ cells, also named Δtether cells (Manford et al, 2012; Hoffmann et al, 2019).

First, we performed control experiments to address whether and how the Tcbs may be activated in the scs2/22Δ ist2Δ triple mutant cells. The E-Syt1 protein has been shown to be activated upon increases in cytoplasmic Ca$^{2+}$ (Fernández-Busnadiego et al, 2015; Idevall-Hagren et al, 2015; Bian et al, 2018). To address whether the Tcb-mediated ER–PM contacts formed in scs2/22Δ ist2Δ cells (Collado et al, 2019) correlate with increases in cytoplasmic Ca$^{2+}$, we examined the cytoplasmic Ca$^{2+}$ reporter GCaMP3 in scs2/22Δ ist2Δ cells. Consistent with Tcb protein activation, GCaMP3 fluorescence was increased in scs2/22Δ ist2Δ cells compared to wild-type cells, as measured by high-content quantitative flow cytometry assays (twofold, Fig 1B) and fluorescence microscopy (1.2-fold, Fig S1A). Furthermore, impairment of Ca$^{2+}$ influx in the scs2/22Δ ist2Δ mutant cells phenocopied loss of the Tcb proteins in scs2/22Δ ist2Δ cells. Specifically, loss of the stress-activated Ca$^{2+}$ channel Mid1 (Iida et al, 1994) in the scs2/22Δ ist2Δ mutant cells (scs2/22Δ ist2Δ mid1Δ) conferred increased resistance to the drug myriocin, similar to the Δtether mutant cells (Fig S1B). Thus, cytoplasmic Ca$^{2+}$ is elevated in scs2/22Δ ist2Δ cells, consistent with a previous study (Kato et al, 2017), and the Mid1-dependent high-affinity Ca$^{2+}$ influx system (termed HACS) (Muller et al, 2001) may contribute to Tcb protein function.

Following these control experiments, we investigated roles of the Tcb proteins in phosphoinositide metabolism at the PM using quantitative microscopy and mass spectrometry-based lipidomics. PI(4,5)P$_2$ and its precursor PI4P (phosphatidylinositol 4-phosphate) are the two major phosphoinositide species present at the PM in yeast and are generated through sequential phosphorylation of PI at the PM (Figs 1C and S1C). Previous studies have implicated metazoan E-Syt family members in the transfer of PI from the ER to the PM during the phosphoinositide cycle (Chang et al, 2013; Saheki et al, 2016; Nath et al, 2020). Yet these studies relied on PI(4,5)P$_2$ synthesis as a proxy for PI transfer, and they did not monitor PI4P synthesis which would be directly impacted if transfer of PI from the ER to the PM was impaired. To address roles of the Tcb proteins in phosphoinositide metabolism at the PM, we first examined the localisation of PI4P and PI(4,5)P$_2$ biosensor FLAREs (fluorescent lipid-associated reporters), GFP-P4C and GFP-2xPH$_{PLC\delta}$, respectively.

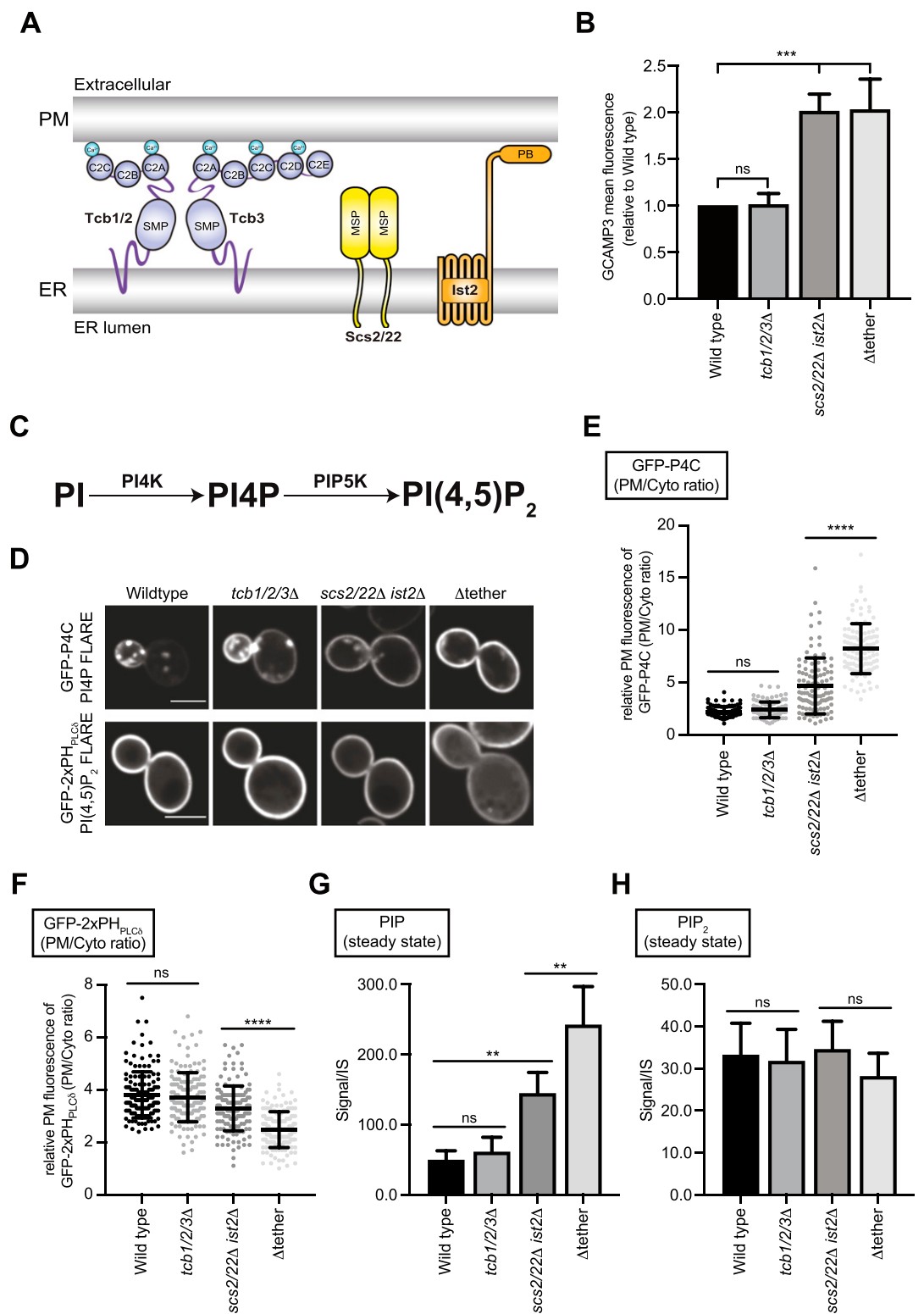

**Figure 1. The tricalbins do not regulate phosphatidylinositol production in the ER but influence PI(4,5)P$_2$ homeostasis at the plasma membrane (PM).**
**(A)** Schematic representations of ER–PM tethering proteins found in budding yeast. These include the tricalbins (Tcb), Scs2 and Scs22, and Ist2 proteins. This study is focused on elucidating specific roles of the Tcb proteins in PM homeostasis. Abbreviations: SMP, synaptotagmin-like mitochondrial lipid–binding domain; C2, C2 domain; MSP, Major sperm protein domain; PB, Polybasic stretch. **(B)** Mean fluorescence intensity of cytoplasmic GCaMP3 reporter of wild-type, *tcb1/2/3Δ*, *scs2/22Δ ist2Δ*, and Δtether cells as measured by flow cytometry (100,000 cells measured per experiment). Data represent mean ± SD from three independent experiments. ***$P > 0.001$.
**(C)** Schematic representation of PI4P and PI(4,5)$_2$ production at the PM and the kinases involved. **(D)** PI4P (GFP-P4C) and PI(4,5)P$_2$ (GFP-2xPH$_{PLCδ}$) FLARE localisation in wild

Localisation of the PI4P and the PI(4,5)P$_2$ biosensor FLAREs were not affected by loss of the Tcb proteins alone (*tcb1/2/3Δ* cells), as compared to wild-type control cells (Fig 1D–F). In contrast, there was a significant increase in the localisation of the PI4P FLARE (GFP-P4C) at the PM in *scs2/22Δ ist2Δ* cells, as compared with wild-type cells (Fig 1D and E), consistent with previous reports. Previous studies have shown that Scs2/22 and Ist2 recruit the PI4P exchange proteins Osh2, Osh3, Osh6, and Osh7 to ER-PM contacts (Loewen & Levine, 2005; D'Ambrosio et al, 2020) and that loss of Scs2/22 and Ist2 results in increased PI4P levels (Manford et al, 2012). Because Tcb-mediated ER–PM contacts are induced in *scs2/22Δ ist2Δ* cells (Collado et al, 2019), the Tcb proteins may contribute to the increase in PI4P at the PM in these cells (e.g., by facilitating PI transfer to the PM as proposed for metazoan E-Syt family members). If this is the case, then levels of PI4P at the PM should be lower in Δtether cells (that lack the Tcb proteins) as compared with *scs2/22Δ ist2Δ* cells (that express the Tcb proteins). However, contrary to this model, the PI4P FLARE was significantly increased at the PM of Δtether cells compared with *scs2/22Δ ist2Δ* cells (Fig 1D and E), suggesting that loss of the Tcb proteins does not impact PI synthesis in the ER, delivery of PI to the PM, or PI4P generation at the PM. We also monitored the distribution of the PI(4,5)P$_2$ FLARE in *scs2/22Δ ist2Δ* and Δtether cells. Whereas there was an increase in the PI4P FLARE signal at the PM of Δtether cells compared with *scs2/22Δ ist2Δ* cells, there was a significant decrease in the relative intensity of the PI(4,5)P$_2$ FLARE at the PM in Δtether cells compared with *scs2/22Δ ist2Δ* cells (Fig 1D and F). Thus, although the Tcb proteins are not required for PI4P generation at the PM, they may indirectly contribute to PI(4,5)P$_2$ homeostasis at the PM.

Next, we confirmed the PI4P and PI(4,5)P$_2$ FLARE results using quantitative lipidomics. For these experiments, we measured levels of phosphoinositide species in ER–PM tether mutants by liquid chromatography–electrospray ionization-tandem mass spectrometry (LC-ESI-MS/MS) analysis (Clark et al, 2011). Levels of phosphatidylinositol (PI), phosphatidylinositol phosphate (PIP), and phosphatidylinositol bis-phosphate (PIP$_2$) species analysed in each of the strains are reported in Table S1. Consistent with the microscopy results, levels of mono-unsaturated phosphatidylinositol phosphate (PIP), which constitute the majority of PI4P species generated at the PM (Wenk et al, 2003; Nishimura et al, 2019), progressively increased between wild-type cells, *scs2/22Δ ist2Δ* cells, and Δtether cells, with the Δtether cells containing significantly higher PIP steady-state levels than the *scs2/22Δ ist2Δ* cells (Figs 1G and S1C). In contrast, none of the strains tested showed significant changes in steady-state levels of any PI species (Fig S1C), indicating that PI metabolism is unaltered in cells lacking the Tcb proteins. Moreover, despite the significant increase in PIP steady-state levels in the Δtether cells, the total level of phosphatidylinositol bis-phosphate (PIP$_2$) was not increased (Figs 1G and H and S1C). The Δtether cells even displayed decreased steady-state

levels of mono-unsaturated PIP$_2$ species (34:1 and 36:1) as compared with *scs2/22Δ ist2Δ* cells (Fig S1C). As expected, no significant differences in PIP or PIP$_2$ levels were detected between *tcb1/2/3Δ* cells and wild type control cells (Figs 1 and S1C). Thus, although previous studies have implicated metazoan E-Syt family members in PI transfer during the phosphoinositide cycle (Chang et al, 2013; Saheki et al, 2016; Nath et al, 2020), the budding yeast Tcb proteins are not required for PI synthesis in the ER, delivery of PI to the PM, or PI4P generation at the PM. Instead, the Tcbs may be indirectly involved in the conversion of PI4P to PI(4,5)P$_2$ at the PM in a manner that is independent of PI transport and PI4P synthesis. Importantly, these results point out Tcb functions that are different from previously proposed roles of metazoan E-Syt family members.

## The tricalbins regulate phosphatidylserine homeostasis at the PM

We next investigated whether the Tcb proteins regulate the homeostasis of other phospholipids at the PM. Intriguingly, whereas the total level of PI(4,5)P$_2$ was not significantly reduced in the Δtether cells as compared with wild-type cells (Fig 1H), the PI(4,5)P$_2$ FLARE was significantly reduced at the PM in the Δtether cells (Fig 1F). Previous work has indicated that the PI(4,5)P$_2$ FLARE (using the PH domain from phospholipase C$_\delta$) preferentially detects PI(4,5)P$_2$ in membranes that also contain phosphatidylserine (PS) (Nishimura et al, 2019). PS is enriched in the cytosolic leaflet of the PM and contributes to its overall negative charge (Yeung et al, 2008; Fairn et al, 2009; Nishimura et al, 2019). PS is synthesized in the ER via the CDP-DAG pathway in yeast (see Fig 5A) and then either transferred to the PM via vesicular and non-vesicular transport pathways or converted to other phospholipids (Figs 2A and S2B) or. The generation of Tcb-mediated ER peaks facing the PM has been suggested to facilitate non-vesicular lipid transport from the ER to the PM (Collado et al, 2019). We therefore considered whether the Tcb proteins facilitate the delivery of PS to the PM. First, we analysed the distribution of a PS FLARE (GFP-Lact-C2) (Fairn et al, 2009; Nishimura et al, 2019) in the same series of ER–PM tether mutants using quantitative microscopy. Under normal growth conditions, loss of the Tcb proteins did not significantly affect the localisation of GFP-Lact-C2 at the PM, as compared with wild-type control cells (Fig 2B and C). However, there was a significant decrease in the relative intensity of the PS FLARE at the PM in Δtether cells as compared with wild-type and *scs2/22Δ ist2Δ* cells (Fig 2B and C). Notably, the PS FLARE was readily observed on intracellular membrane compartments in the Δtether cells (Fig 2B), consistent with impaired PS transport from the ER to the PM. The relative intensity of the PS FLARE at the PM was also significantly decreased in *scs2/22Δ ist2Δ mid1Δ* cells as compared with *scs2/22Δ ist2Δ* cells (Fig S2A), consistent with a role of the Ca$^{2+}$-activated Tcb proteins in PS homeostasis at the PM.

---

type, *tcb1/2/3Δ*, *scs2/22Δ ist2Δ*, and Δtether cells. Scale bars, 4 $\mu$m. **(E)** Quantitation of GFP-P4C intensity at the PM of the mother cell. Data represent mean ± SD. Total number of cells analysed in three independent experiments: wild type n = 104, *tcb1/2/3Δ* n = 102, *scs2/22Δ ist2Δ* n = 107, Δtether n = 105. ****$P$ > 0.0001. **(F)** Quantitation of GFP-2xPH$_{PLC\delta}$ intensity at the PM of the mother cell. Data represent mean ± SD. Total number of cells analysed in three independent experiments: wild type n = 146, *tcb1/2/3Δ* n = 149, *scs2/22Δ ist2Δ* n = 146, Δtether n = 147. ****$P$ > 0.0001. **(G, H)** Lipidomic analysis of total PIP and PIP2 species in wild-type, *tcb1/2/3Δ*, *scs2/22Δ ist2Δ*, and Δtether cells. Data represent mean ± SD (N = 5). **$P$ > 0.01; ns, not significant. Also see Fig S1.
Source data are available for this figure.

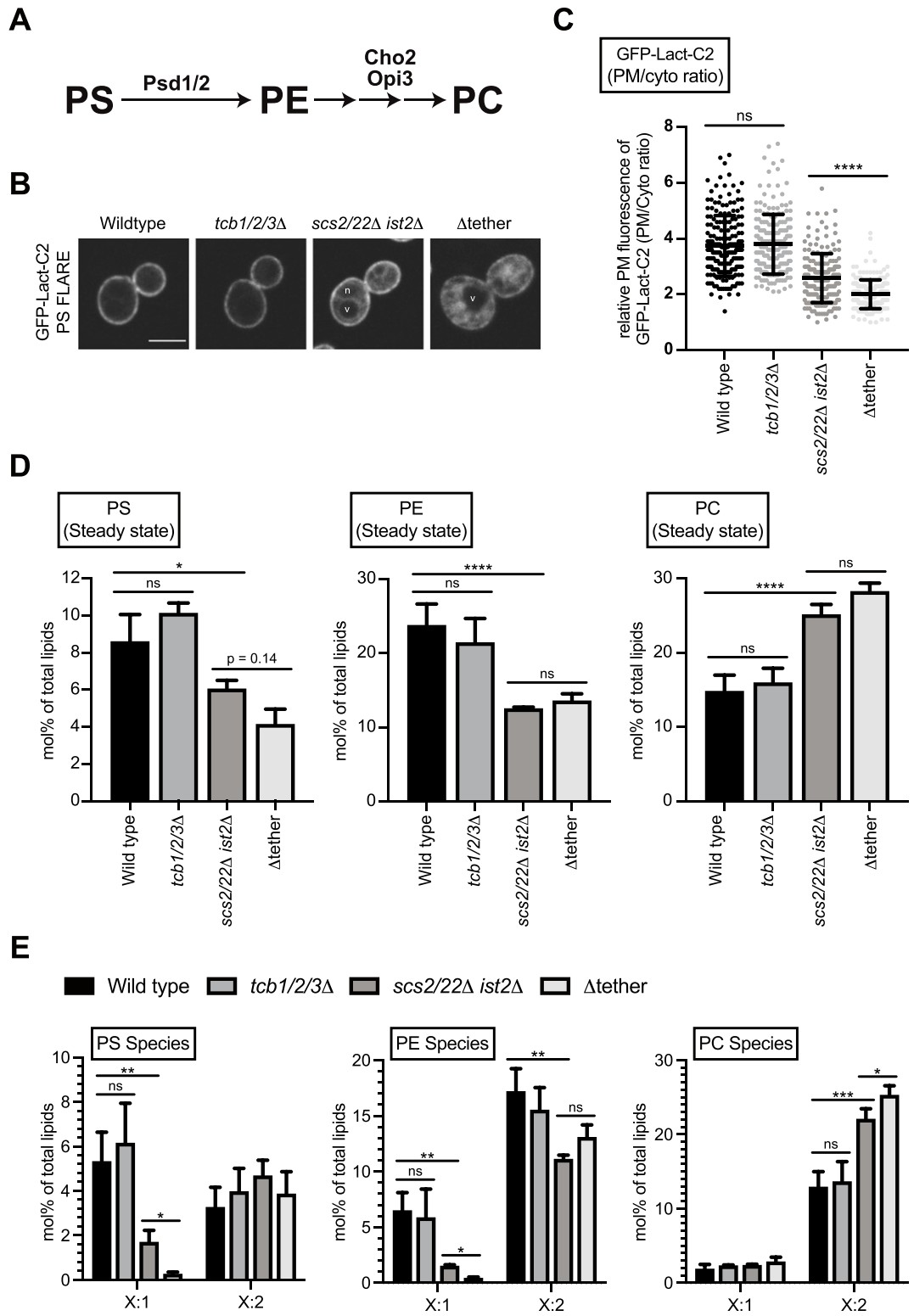

**Figure 2. The tricalbins regulate phosphatidylserine homeostasis at the plasma membrane.**
**(A)** Schematic representation of PS, PE, and PC production and the enzymes involved. **(B)** PS FLARE (GFP-Lact-C2) localisation in wild type, *tcb1/2/3Δ*, *scs2/22Δ ist2Δ*, and Δtether cells. Scale bars, 4 μm. **(C)** Quantitation of relative GFP-Lact-C2 intensity at the plasma membrane of the mother cell. Data represent mean ± SD. Total number of cells analysed in three independent experiments: wild type n = 196, *tcb1/2/3Δ* n = 194, *scs2/22Δ ist2Δ* n = 187, Δtether n = 184. ****$P > 0.0001$. **(D)** Lipidomic analysis of total PS, PE and PC species in wild type, *tcb1/2/3Δ*, *scs2/22Δ ist2Δ*, and Δtether cells. Data represent mean ± SD (n = 3). *$P > 0.1$; ns, not significant. **(E)** The fatty acid degree of

The roles of ER–PM contacts in phospholipid regulation remain incompletely understood and are even contentious and debated. Two previous studies have argued that ER–PM contacts in yeast are primarily involved in lipid metabolism, rather than lipid transfer between the ER and PM (Quon et al, 2018; Wang et al, 2020). In contrast, more recent studies have proposed that the Osh6 and Osh7 proteins bind Ist2 at ER–PM contacts where they transport PS from the ER to the PM (D'Ambrosio et al, 2020; Wong et al, 2021). Moreover, previous studies have indicated that non-vesicular lipid transport and metabolism are co-regulated in yeast (Maeda et al, 2013; Kannan et al, 2017; D'Ambrosio et al, 2020; Wong et al, 2021). To address these issues in further detail, we investigated roles of the Scs2/22, Ist2, and Tcb1/2/3 proteins in lipid homeostasis using quantitative lipidomics. Levels of the lipids analysed (glycerolipids, phospholipids, and neutral lipids) in each of the strains are reported in Table S2. Consistent with the microscopy results, steady-state levels of PS were reduced in the scs2/22Δ ist2Δ and Δtether mutant cells, but not in tcb1/2/3Δ cells, as compared with wild-type control cells (Figs 2D and S2B). PS steady-state levels were slightly lower in the Δtether cells as compared with scs2/22Δ ist2Δ cells, but this change was not statistically significant (Fig 2D). Although overall PS levels were not significantly changed upon loss of the Tcb proteins, species-level analyses clearly revealed changes in acyl chain saturation. Loss of the Tcb proteins in the scs2/22Δ ist2Δ background resulted in a significant decrease in mono-unsaturated PS species (threefold; designated as X:1, Fig 2E). In particular, mono-unsaturated 34:1 PS, which is the major PS species enriched at the PM (Schneiter et al, 1999), was significantly decreased in Δtether cells compared with scs2/22Δ ist2Δ cells (>7-fold; Fig S2B). In contrast, there were no significant differences in the levels of di-unsaturated PS species in any of the strains examined (designated as X:2, Figs 2E and S2B). Thus, PS synthesis per se was not completely disrupted in the Δtether cells. Consistent with this, cho1Δ mutant cells defective in PS synthesis rely upon exogenous ethanolamine or choline to support PE and PC synthesis for growth (Atkinson et al, 1980), but the Δtether cells are viable in the absence of exogenous ethanolamine and choline and grow on standard media (see Fig S1) (Hoffmann et al, 2019). Thus, although levels of "ER-like" di-unsaturated PS species are not affected upon loss of Scs2/22, Ist2, and Tcb1/2/3, loss of these proteins resulted in specific and additive decreases in mono-unsaturated 34:1 PS species that are enriched in the cytoplasmic leaflet of the PM, consistent with impaired transport of newly synthesized mono-unsaturated 34:1 PS from the ER.

PS can be converted to phosphatidylethanolamine (PE) through decarboxylation reactions carried out by Psd1 (in the ER and mitochondria) (Friedman et al, 2018) and Psd2 (in late Golgi/early endosomal compartments), and PE can be methylated by Cho2 and Opi3 to generate phosphatidylcholine (PC) (Figs 2A and S2B). Loss of Psd1 and Psd2 is lethal in the absence of ethanolamine or choline supplementation needed to support the Kennedy phospholipid synthesis pathway (Maeda et al, 2013; Wang et al, 2020; Wong et al,

2021). A recent study found that loss of Psd1 (shown to function at the ER and mitochondria) is synthetic lethal in the Δtether cells in the absence of ethanolamine supplementation (Wong et al, 2021). The finding that Δtether cells are reliant on Psd1 indicates that (1) Psd1 function remains intact in the ER-mitochondrial system(s) in the Δtether cells, (2) Psd2 activity at late Golgi and endosomal compartments is impaired, and (3) that PS species may be converted to other phospholipids in the Δtether cells. Accordingly, the quantitative lipidomic analyses in this study revealed several changes in PE and PC species in the mutant cells. Steady-state levels of PE were reduced in the scs2/22Δ ist2Δ and Δtether mutant cells, but not in tcb1/2/3Δ cells, as compared with wild-type control cells (Fig 2D). Notably, mono-unsaturated 32:1 and 34:1 PE, the major PE species enriched at the PM (Schneiter et al, 1999), were specifically decreased in the scs2/22Δ ist2Δ and Δtether mutant cells as compared with wild-type cells (Figs 2E and S2B). Mono-unsaturated 34:1 PE levels were decreased even further in the Δtether cells as compared with scs2/22Δ ist2Δ cells (fourfold; Figs 2E and S2B). In contrast, there were no significant differences in the levels of di-unsaturated 34:2 PE in any of the strains examined (Figs 2E and S2B). Whereas steady-state levels of PS and PE were reduced in the scs2/22Δ ist2Δ and Δtether mutant cells, steady-state levels of PC increased in turn (Fig 2D). This was specifically due to significant increases in di-unsaturated forms of PC (32:2, 34:2, and 36:2 PC) in the scs2/22Δ ist2Δ and Δtether mutant cells as compared with wild-type cells (Figs 2E and S2B). Moreover, di-unsaturated PC species were increased in the Δtether cells as compared with scs2/22Δ ist2Δ cells (Figs 2E and S2B). Altogether, these results suggest that delivery of PS (and PE) from the ER is impaired in the absence of Scs2/22, Ist2, and the Tcbs. Consequently, mono-unsaturated PS and PE species retained in the ER may be desaturated and converted to di-unsaturated PC species that accumulate in the mutant cells. The data also reveal a potential role of the Tcb proteins in 34:1 PS and 34:1 PE homeostasis upon loss of Scs2/22 and Ist2 function. Altogether, the species-level (acyl chain) lipidomics data suggest that ER-PM contact site proteins regulate PS transport and metabolism, providing strong support for previous studies that have suggested this notion (Maeda et al, 2013; D'Ambrosio et al, 2020; Wong et al, 2021).

## Tricalbins regulate phosphatidylserine distribution upon heat stress conditions

To better understand the primary roles of the Tcb proteins, we next focused on characterisation of the tcb1/2/3Δ cells, rather than the Δtether cells that lack multiple proteins (the Tcbs, Scs2/22, and Ist2). The Tcb proteins are necessary for the formation of heat-induced ER-PM contacts as well as the maintenance of PM integrity upon brief heat stress (10 min 42°C) (Collado et al, 2019). However, it is not understood how the Tcb proteins maintain PM integrity upon heat stress. In control experiments, a brief heat stress (10 min 42°C) induced heterogeneous transient cytoplasmic Ca²⁺ bursts approximately fivefold higher than basal levels at 26°C, as detected by

unsaturation for PS, PE, and PC species in wild-type, tcb1/2/3Δ, scs2/22Δ ist2Δ, and Δtether cells. Data represent mean ± SD (n = 3). ***P > 0.001, **P > 0.01, *P > 0.1; ns, not significant. Also see Fig S2.
Source data are available for this figure.

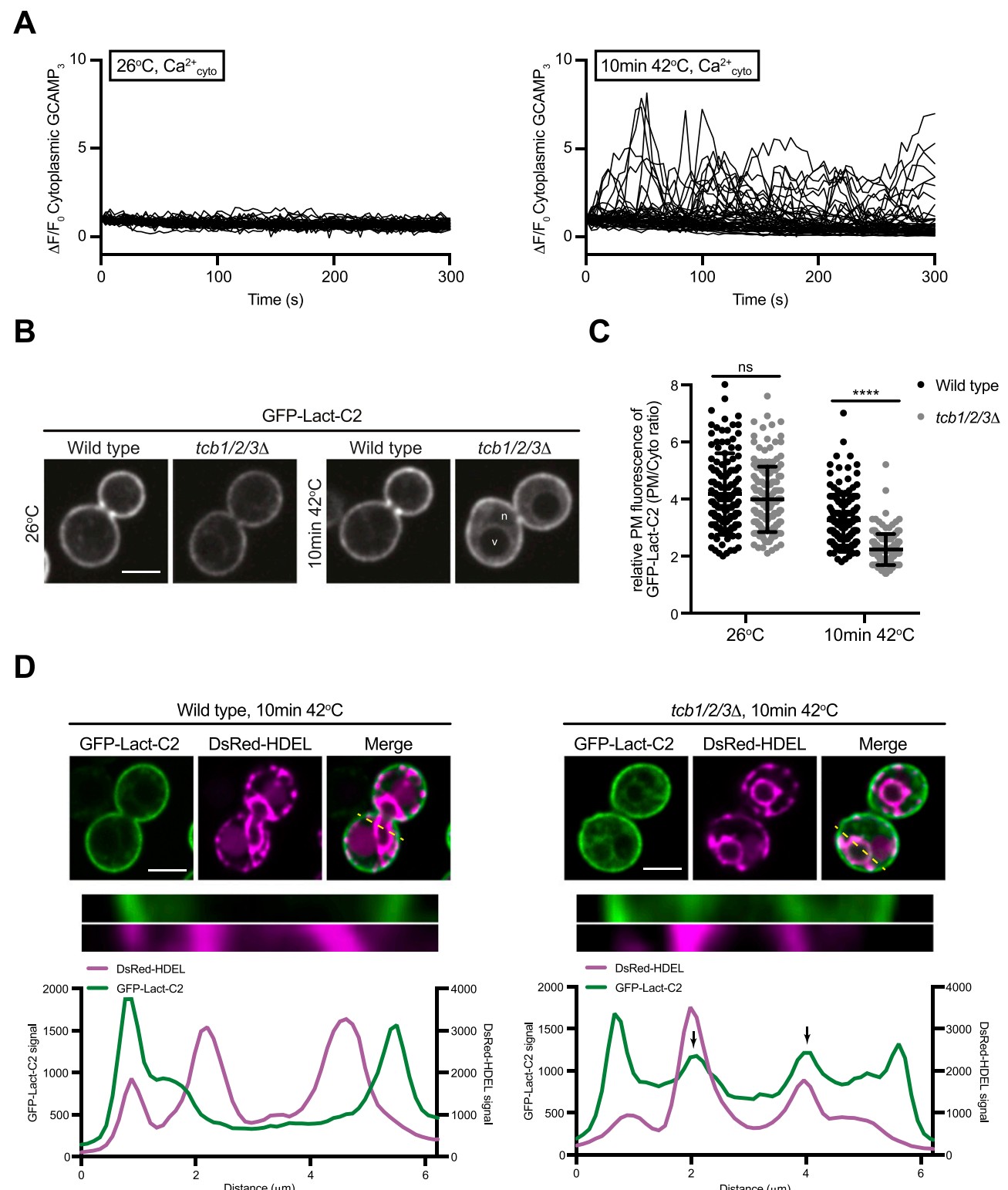

**Figure 3. The tricalbins control phosphatidylserine distribution upon heat stress conditions.**
**(A)** Normalised fluorescence ($\Delta F/F_0$) traces of wild-type cells expressing a cytoplasmic $Ca^{2+}$ sensor, GCAMP3 imaged at 26°C or following a 10-min incubation at 42°C. Each line represents an individual cell, 49 (top) and 48 (bottom) traces for each condition. **(B)** PS FLARE (GFP-Lact-C2) localisation in wild-type and *tcb1/2/3Δ* cells at 26°C or after 10 min at 42°C. Scale bar, 4 *μm*. **(C)** Quantitation of relative GFP-Lact-C2 levels at the plasma membrane at 26°C and after 10 min at 42°C in wild-type and *tcb1/2/3Δ* cells. Data represent mean ± SD. Total number of cells analysed in three independent experiments: wild type 26°C n = 153, wild type 10 min 42°C n = 152, *tcb1/2/3Δ* 26°C n = 152, *tcb1/2/3Δ* 10 min 42°C n = 151 cells. ****$P$ > 0.0001. **(D)** GFP-Lact-C2 localisation in wild type (left) and *tcb1/2/3Δ* (right) cells, co-expressing the ER marker DsRed-HDEL

GCaMP3 fluorescence (Figs 3A and S3A–C). We next addressed whether the Tcb proteins regulate PS distribution in response to heat stress. Remarkably, whereas wild-type cells showed only a small drop in relative levels of the PS FLARE at the PM after a brief shift from 26°C to 42°C, PM localisation of the PS FLARE significantly decreased in the tcb1/2/3Δ mutant cells at 42°C (Fig 3B and C). Moreover, the PS FLARE localised diffusely in the cytoplasm and accumulated on intracellular membrane compartments at 42°C in the tcb1/2/3Δ mutant cells (Fig 3B), suggesting that the Tcb proteins are involved in the regulation of PS at the PM upon heat stress. To identify the intracellular compartments on which PS accumulates in tcb1/2/3Δ cells, we monitored PS FLARE co-localisation with markers for the ER (DsRed-HDEL) or vacuole membrane (FM4-64). Although the PS FLARE did not localise at the nuclear ER or vacuole membrane in wild-type cells at 42°C (Figs 3D and S3D), the PS reporter localisation significantly increased at both the ER and vacuole membrane in tcb1/2/3Δ mutant cells at 42°C (Figs 3D and S3D–F). Thus, the Tcb proteins regulate the distribution of PS upon heat stress conditions.

### ER-localised Tcb3 associates with Sfk1 at the PM

Sfk1 is an integral PM protein that has been implicated in PM phospholipid asymmetry and PM integrity (Mioka et al, 2018; Kishimoto et al, 2021), as well as heat-induced PI(4,5)P$_2$ synthesis (Audhya & Emr, 2002). We observed that the ER-localised Tcb3 protein is in close proximity to the integral PM protein Sfk1, as assessed by split GFP bimolecular fluorescence complementation (BiFC) assays (Fig 4A and B). Intriguingly, the Tcb3-Sfk1 association was enriched at the cortex of mother cells as compared with daughter cells (Fig 4A and B). In control experiments, Tcb3 split GFP fusion proteins formed cortical patches dependent on the Tcb3 C2 domains (Fig S4A) that target Tcb3 to the cortical ER (Manford et al, 2012). In addition, Sfk1 and Tcb3 did not associate with the ER membrane protein Sec61 in split GFP BiFC assays (Fig S4A), indicating a specific association between Sfk1 and Tcb3. Consistent with this, the C-terminal cytoplasmic domain of Sfk1 was required for efficient association with Tcb3, as the split GFP signal intensity significantly decreased in cells expressing a truncated version of Sfk1 (Sfk1$^{Δ286-353}$) (Fig 4B). Sfk1 also associated with Tcb1 and Tcb2 as assessed by split GFP BiFC assays (Fig S4B), consistent with previous work suggesting that Tcb3 may form and function in heterodimers with Tcb1 and Tcb2 (Manford et al, 2012; Collado et al, 2019). Similar to Tcb3-Sfk1 assemblies, cortical Tcb1/2-Sfk1 assemblies were enriched in mother cells as compared with daughter cells (Fig S4B). As mentioned, Sfk1 has been implicated in PM lipid organisation and integrity (Mioka et al, 2018; Kishimoto et al, 2021). Tcb3 has also been shown to be necessary for maintenance of PM integrity upon heat stress conditions (Collado et al, 2019). We therefore assessed the localisation of Tcb3-GFP with Sfk1-mCherry after a brief heat stress (10 min 42°C). Sfk1 and Tcb3 displayed increased co-

localisation at the cell cortex upon a brief incubation at 42°C (Fig 4C and D). Thus, Tcb3 is localised in proximity to Sfk1 at the PM, especially in response to heat stress conditions that elevate cytoplasmic Ca$^{2+}$ signalling and induce Tcb3-mediated ER–PM tethering.

To address the functional significance of the Tcb–Sfk1 association, we examined whether loss of Tcb and Sfk1 function result in similar phenotypes. Accordingly, PM localisation of the PS FLARE was decreased upon a brief shift to 42°C upon loss of Sfk1 or truncation of the Sfk1 cytoplasmic tail (Fig 4E), similar to tcb1/2/3Δ mutant cells (Fig 3C). Sfk1 has recently been shown to regulate PM sterol organisation (Kishimoto et al, 2021). In reciprocal experiments, we addressed whether the Tcb proteins also regulate sterol availability at the PM (Fig 4F). In wild-type cells, the sterol probe mCherry-D4H (Maekawa & Fairn, 2015; Marek et al, 2020) localised primarily to the daughter bud with relatively low levels observed at the PM of the mother cell (Fig 4F), consistent with previous reports (Marek et al, 2020; Kishimoto et al, 2021). However, the distribution of the sterol probe was altered in cells lacking the Tcb proteins, with increased mCherry-D4H signal at the PM of the mother cell (~3-fold greater than wild type control cells; Fig 4F). Thus, Tcb3 and Sfk1 not only co-localise, but they also share common functions in PM lipid organisation.

We further investigated roles of the Tcb proteins and Sfk1 in lipid homeostasis during heat stress using quantitative lipidomics. Levels of the lipids analysed in each of the strains after a brief heat stress (10 min at 42°C) are reported in Table S3. The levels of PS and PE were slightly reduced at 42°C in cells lacking the Tcb proteins compared with wild-type cells (~1.3-fold and ~1.4-fold, respectively, Fig 5B and C), whereas PC levels were not significantly affected (Fig 5D). The decrease in PS may due to product feedback inhibition of the PS synthase Cho1 (Kannan et al, 2017), as the PS FLARE accumulates at the ER in tcb1/2/3Δ cells at at 42°C (Figs 3D and S3E). In line with this, levels of phosphatidic acid (PA), a precursor of PS synthesis (Fig 5A), were higher in tcb1/2/3Δ mutant cells at 42°C compared with wild type (~4-fold, Fig 5E). The increase in PA steady-state levels may also be due to PS-mediated inhibition of the PA phosphatase Pah1 at the ER and vacuole (Dey et al, 2017). Although cells lacking Sfk1 displayed reduced PS FLARE localisation at the PM, there were no significant changes in total PS levels, or any other lipids detected by mass spectrometry, in sfk1Δ cells at 42°C compared with wild-type control cells (Fig 5). Thus, whereas the Tcb proteins are involved in the delivery of PS to the PM upon heat stress, Sfk1 may regulate the transbilayer organisation of PS and other lipids within the PM under these conditions.

A previous study implicated E-Syt1 in recycling DAG from the PM to the ER during the phosphophoinositide cycle (Saheki et al, 2016). However, there was no difference in the levels of DAG between wild-type cells and tcb1/2/3Δ or sfk1Δ mutant cells at either 26°C or 42°C (Fig 5F and Tables S2 and S3). In addition, a DAG FLARE (GFP-C1$_{PKD}$) did not accumulate at the PM in tcb1/2/3Δ mutant cells at 26°C or

---

(magenta), after 10 min at 42°C. Top panels: representative midsection images. Scale bar, 4 μm. Bottom panels: linearised signal (yellow dotted line on midsection images) through the plasma membrane and nER and representative graphs showing the intensity profiles for both channels. Arrows indicate peaks in the GFP-Lact-C2 signal that relate to the nER. Also see Fig S3.
Source data are available for this figure.

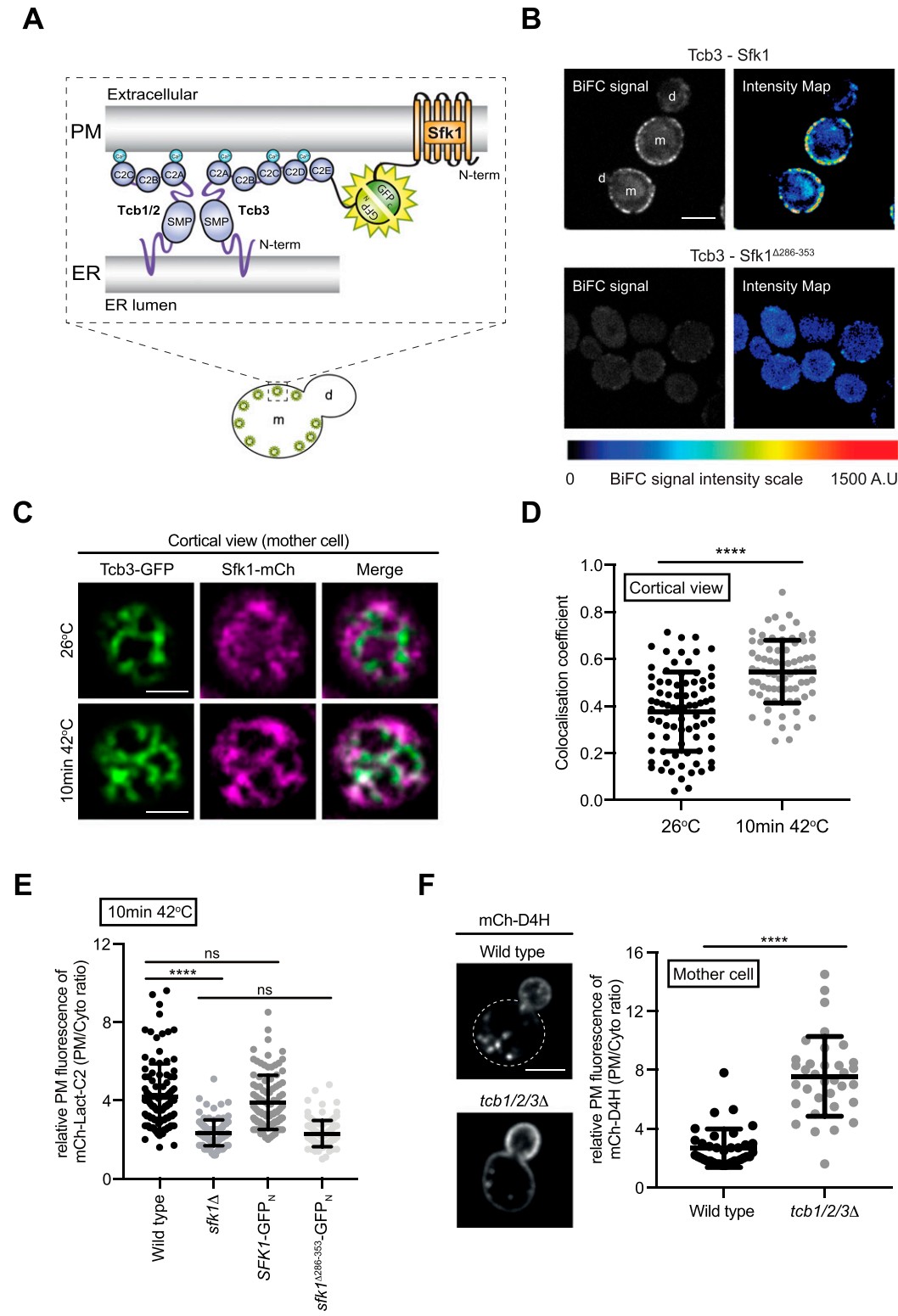

**Figure 4. ER-localised Tcb3 associates with Sfk1 at the plasma membrane (PM).**
**(A)** Cartoon displaying the bi-molecular fluorescence (BiFC) split GFP assay to assess Tcb3-Sfk1 proximity. The N-terminal half of GFP (GFP_N) and the C-terminal half of GFP (GFP_C) form a fluorescent GFP only when their fusion partners, in this case Tcb3 and Sfk1, are in close spatial proximity with each other. m, mother cell; d, daughter cell. **(B)** Tcb3-GFP_N associates with Sfk1-GFP_C but not with a mutant Sfk1 lacking its cytoplasmic C terminus (Sfk1^{Δ286-353}-GFP_C). The pseudo-coloured images indicate the scale of specific BiFC signals (blue, moderate; red, strong). m, mother cell; d, daughter cell. Scale bars, 4 μm. **(C)** Cortical localisation of Tcb3-GFP and Sfk1-mCherry at 26°C or after 10 min at 42°C in the mother cell. Scale bars, 4 μm. **(D)** Quantitation of Tcb3-GFP and Sfk1-mCherry co-localisation (Pearson's coefficient) at 26°C or after 10 min at

42°C (Fig S5A). Moreover, steady-state PI levels were unaffected by loss of Sfk1 or the Tcb proteins at 42°C (Fig 5G), indicating that the CDP-DAG pathway (Fig 5A) remains intact in these cells. Furthermore, the Tcb proteins were not required for PI, PI4P, or $PI(4,5)P_2$ synthesis upon heat stress conditions, as assessed by [3]H-inositol labelling and high-performance liquid chromatography (HPLC) analyses (Fig S5B–D and Table S4). Sfk1 has also been implicated heat-induced $PI(4,5)P_2$ synthesis (Audhya & Emr, 2002). However, the relative levels of the PI(4,5)P2 FLARE at the PM were not significantly different between wild type, sfk1Δ, and tcb1/2/3Δ cells at 26°C or 42°C (Fig S5E and F). Thus, although metazoan E-Syt and TMEM150 family members reportedly function in the phosphoinositide cycle (Chang et al, 2013; Chung et al, 2015; Saheki et al, 2016; Nath et al, 2020), further investigations are needed for Sfk1 and the Tcb proteins.

## The tricalbins regulate Pkc1 localisation under heat stress conditions

As mentioned, the Tcb proteins form heat-induced ER–PM contacts and are required for PM integrity under these conditions (Collado et al, 2019). Yet it is not known how the Tcb proteins contribute to PM integrity. We therefore examined whether the Tcb proteins regulate the distribution of a PS-binding protein necessary for PM integrity. Pkc1 is a serine/threonine kinase (ortholog of mammalian nonconventional protein kinase C family members) that controls cell integrity (Kamada et al, 1995; Roelants et al, 2017; Heinisch & Rodicio, 2018). Under normal growth conditions, Pkc1 localises to sites of polarised growth including the bud tip and bud neck (Andrews & Stark, 2000; Denis & Cyert, 2005). However, Pkc1 undergoes changes in localisation and directs changes in the cytoskeletal and secretory apparatus in response to stress conditions that cause cell membrane damage (Delley & Hall, 1999; Kono et al, 2012). The regulatory region of Pkc1 contains proposed PS-binding domains including a C2 domain (Fig 6A) that controls Pkc1 localisation at the PM, and PS has been shown to be required for Pkc1 function (Denis & Cyert, 2005; Dey et al, 2017; Nomura et al, 2017; Roelants et al, 2017; Heinisch & Rodicio, 2018). We examined the localisation of a functional Pkc1-GFP fusion in wild type and tcb1/2/3Δ cells co-expressing mCherry-2xPH$_{PLC\delta}$ as a PM marker. Consistent with previous reports, Pkc1-GFP localised to the bud neck in wild-type cells at 26°C (Fig 6B). Upon a brief heat stress at 42°C, Pkc1-GFP puncta formed at the cortex of both mother and daughter wild-type control cells (Fig 6B–D), along with intracellular puncta corresponding to transport vesicles. In contrast, whereas Pkc1-GFP was recruited to the PM in daughters in tcb1/2/3Δ cells (Fig 6B and C), Pkc1-GFP assemblies were significantly decreased at the cortex of mother cells in tcb1/2/3Δ cells as

compared with wild-type cells at 42°C (~1.5-fold; Fig 6B and D). Heat-induced cortical Pkc1-GFP assemblies were also impaired in mother cells, but not daughter cells, upon loss of Sfk1 (~1.5-fold; Fig S6A), similar to cells lacking the Tcb proteins. Consistent with the specific requirement for Sfk1 and the Tcb proteins in Pkc1 targeting in mother cells, cortical Tcb-Sfk1 split GFP assemblies were more apparent in mother cells than daughter cells (Figs 4B and S4B). Pkc1-GFP also mislocalised to large intracellular puncta in tcb1/2/3Δ cells at both 26°C and 42°C (~60% of cells in both conditions) that are not observed in wild-type cells at either temperature (Fig 6B and E). Thus, the Tcb proteins regulate Pkc1 membrane targeting, particularly recruitment to the PM in mother cells under heat stress conditions.

A previous study has suggested that the Tcb proteins transfer ceramides from the ER to Golgi compartments upon secretory defects (Ikeda et al, 2020). Moreover, de novo sphingolipid synthesis is increased in yeast upon heat stress (Tabuchi et al, 2006; Cowart & Hannun, 2007; Cowart & Obeid, 2007; Omnus et al, 2016). Furthermore, sphingolipids in the extracellular leaflet of the PM are proposed to undergo transbilayer coupling with PS species in the cytoplasmic leaflet of the PM (Sezgin et al, 2017). For all these reasons, we examined whether Tcb-mediated transfer of newly synthesized sphingolipids (long-chain bases and ceramides) from the ER to the PM might be involved in Pkc1 recruitment to the mother cell cortex under heat stress conditions. However, de novo sphingolipid synthesis was not required for heat-induced Pkc1 recruitment to the mother cell cortex (Fig S6B–D). Instead, heat-induced Pkc1 recruitment to the mother cell cortex was enhanced upon pre-treatment with myriocin, a potent inhibitor of sphingolipid synthesis in the ER (Fig S6B–D). Thus, although the Tcb proteins may transfer newly synthesized sphingolipids out of the ER under secretory stress conditions (Ikeda et al, 2020), this activity is not required for the targeting of Pkc1 to the mother cell cortex upon heat stress.

To address whether Pkc1 function becomes limiting upon loss of the Tcb proteins, we examined whether Pkc1 overexpression could rescue PM integrity defects in tcb1/2/3Δ cells. As previously reported (Collado et al, 2019), loss of the Tcb proteins resulted in measurable PM integrity defects upon a brief shift to 42°C, as measured by an established propidium-based PM integrity assay (~15-fold increase in propidium-stained tcb1/2/3Δ cells at 42°C as compared with control cells; Fig 6F). Notably, overexpression of Pkc1 from a high copy plasmid partially rescued the PM integrity defect in tcb1/2/3Δ cells at 42°C (~1.8-fold decrease in propidium-stained tcb1/2/3Δ cells carrying the high copy PKC1 plasmid; Fig 6F). In contrast, expression of a truncated dominant-negative form of Pkc1 (Pkc1-T615) (Parrish et al, 2005) that lacks the C-terminal kinase domain induced PM integrity defects in both wild type and tcb1/2/

---

42°C. Data represent mean ± SD. Total number of cells analysed in three independent experiments: wild type n = 82, tcb1/2/3Δ n = 75. ****P > 0.0001. **(E)** Quantitation of relative mCh-Lact-C2 levels at the PM after 10 min at 42°C in wild type, sfk1Δ, Sfk1-GFP$_N$, Sfk1$^{Δ286-353}$-GFP$_N$ cells. Data represent mean ± SD. Total number of cells analysed in three independent experiments: wild type n = 106, sfk1Δ n = 99, Sfk1-GFP$_N$ n = 96 cells, Sfk1$^{Δ286-353}$-GFP$_N$ n = 105 cells. ****P > 0.0001; ns, not significant. **(F)** Sterol FLARE (mCh-D4H) localisation in wild-type and tcb1/2/3Δ cells. Left panel: representative midsection images. Mother cell PM of wild-type cells is indicated by a dotted line. Scale bar, 4 μm. Right panel: quantitation of relative mCh-D4H levels at the PM in wild type and tcb1/2/3Δ cells. Data represent mean ± SD. Total number of cells analysed in three independent experiments: wild type n = 36, tcb1/2/3Δ n = 34 cells. ****P > 0.0001. Also see Fig S4.
Source data are available for this figure.

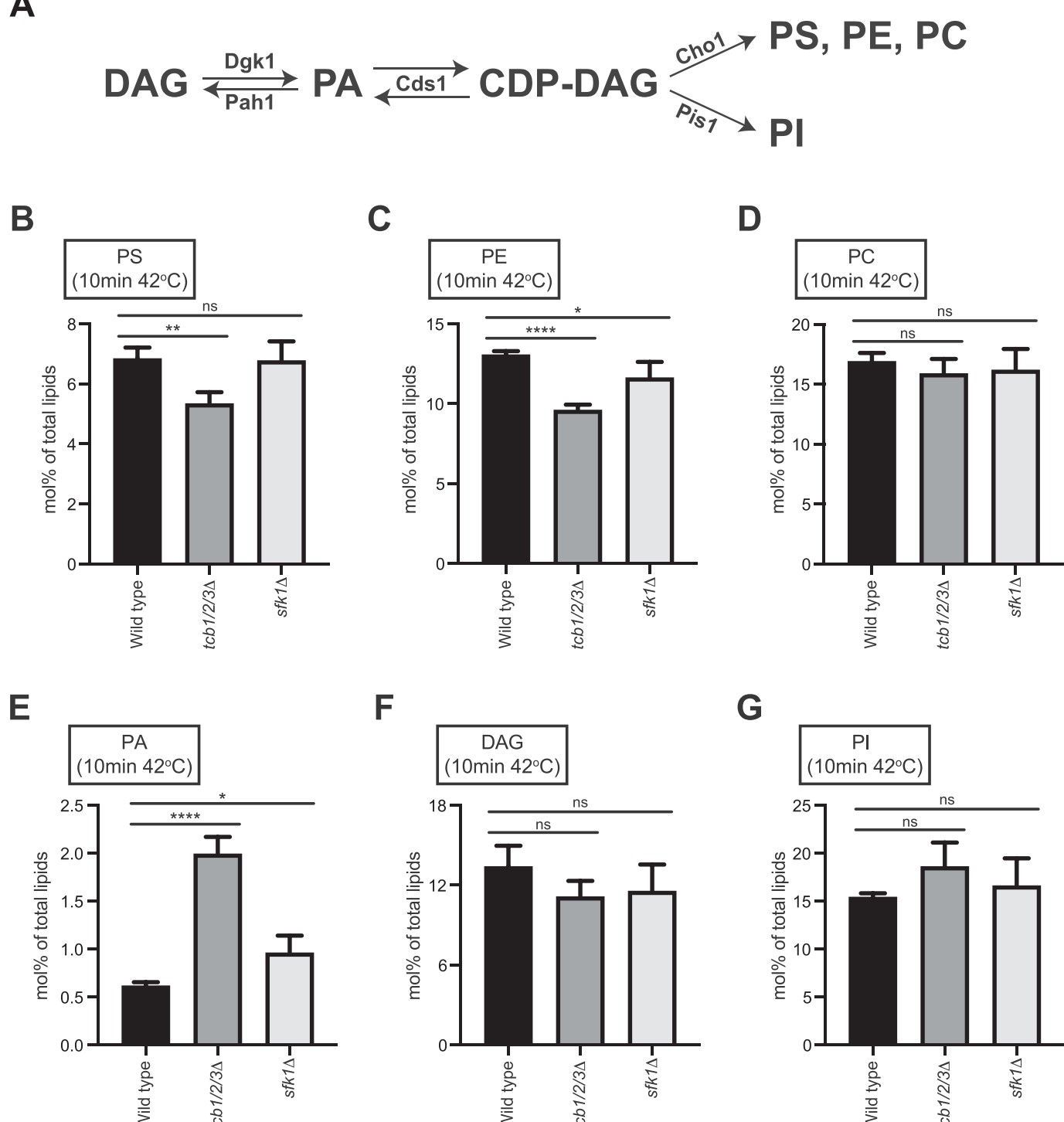

**Figure 5. The PI cycle remains intact upon heat stress in cells lacking the tricalbins.**
**(A)** Schematic representation of PS and PI production from PA via the CDP-DAG pathway and the enzymes involved. **(B, D, E, F, G)** Lipidomic analysis of total PS, PE, PC, PA, DAG, and PI species, respectively, in wild type, *tcb1/2/3Δ* and *sfk1Δ* cells following a 10-min incubation at 42°C. Data represent mean ± SD (n = 3). \*\*\*P > 0.001, \*\*P > 0.01, \*P > 0.1; ns, not significant. Also see Fig S5.

3Δ cells at 26°C (Fig 6F). However, expression of Pkc1-T615 did not exacerbate the PM integrity defects in *tcb1/2/3Δ* cells at 42°C (Fig 6F), further indicating that impaired Pkc1 function contributes to

the heat-induced PM integrity defect in *tcb1/2/3Δ* cells. Therefore, the Tcb proteins maintain PS homeostasis at the PM upon heat stress (Fig 3), and they also direct the PM targeting and activity of

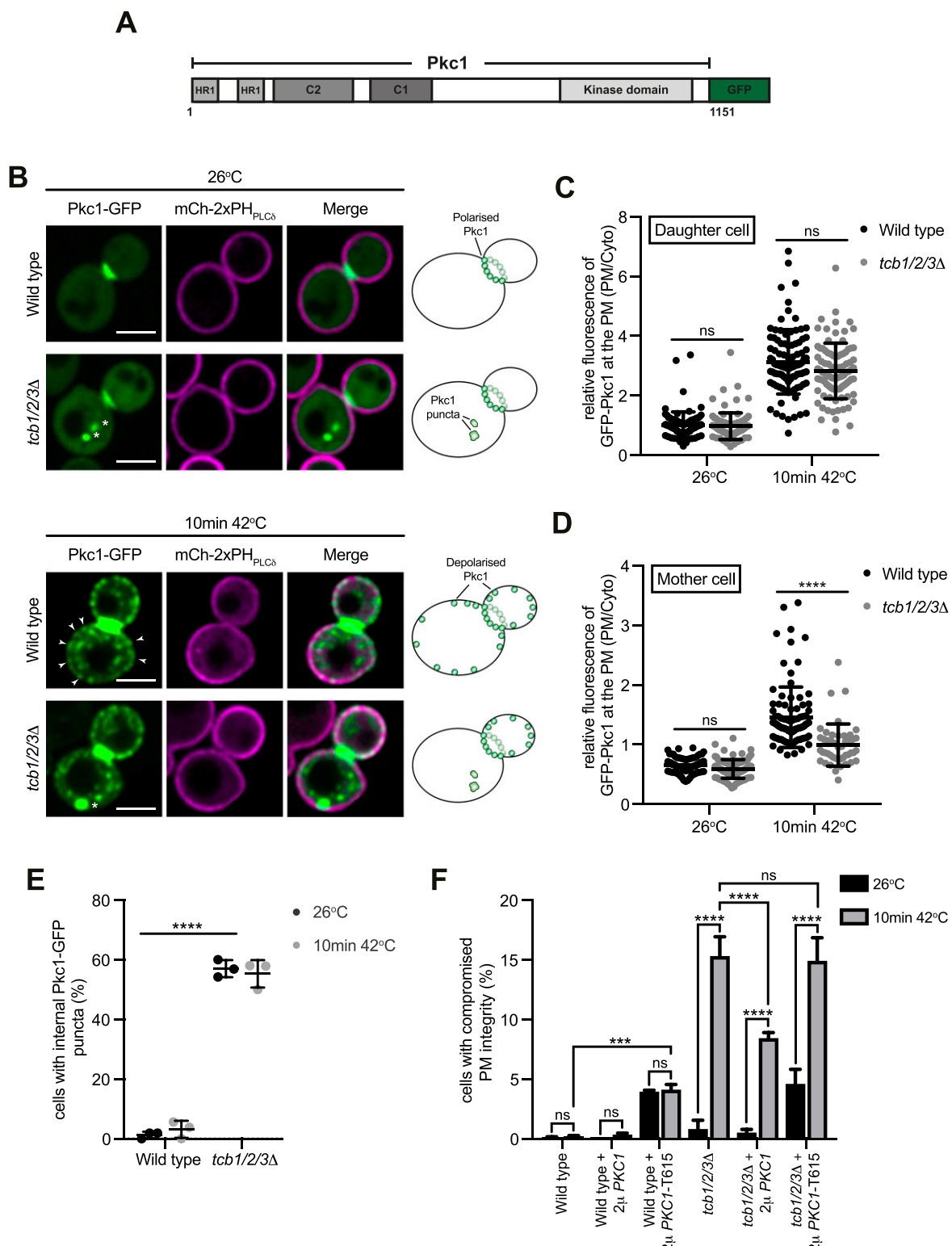

**Figure 6. The tricalbins regulate Pkc1 localisation under heat stress conditions.**
**(A)** Schematic representation of Pkc1-GFP. HR1, putative rho-binding repeat; C2, putative $Ca^{2+}$/lipid-binding motif; Ps, putative pseudosubstrate motif; C1, putative diacylglycerol binding motif. **(B)** Pkc1-GFP (green) localisation in wild type and tcb1/2/3Δ mutant cells, co-expressing the plasma membrane (PM) marker mCh-2xPH$_{PLC\delta}$ (magenta), at 26°C (top panel) and following a brief heat stress (10 min at 42°C) (bottom panel). Arrows indicate recruitment of Pkc1-GFP to the PM of the mother cell. Asterisks indicate internal Pkc1-GFP puncta observed in the tcb1/2/3Δ mutants. Scale bar, 4 μm. **(C, D)** Quantitation of relative Pkc1-GFP levels at the PM of the mother cell (C) and daughter bud (D), in wild-type and tcb1/2/3Δ cells at 26°C or after 10 min at 42°C. Data represent mean ± SD. Total number of cells analysed in four

the PS effector protein Pkc1 that is critical for cellular integrity under these conditions (Fig 6B, D, and F).

### The Osh6 and Osh7 phosphatidylserine transfer proteins are not required for PM integrity upon heat stress conditions

Roles of the Osh6 and Osh7 and their binding partner Ist2 in PS transport have been extensively studied (Maeda et al, 2013; Moser von Filseck et al, 2015; D'Ambrosio et al, 2020; Wong et al, 2021). It has even been suggested that non-vesicular PS transfer is specifically carried out by the Osh6/7 proteins and that other tether/transfer proteins do not carry out this function (Wong et al, 2021). However, we speculated that the Tcb proteins may regulate PS distribution under stress conditions in which the Ist2-mediated Osh6/7 system is attenuated. We compared the relative roles of ER–PM tether proteins and lipid transfer proteins in PM homeostasis upon heat stress. Notably, PM levels of the PS FLARE in *tcb1/2/3Δ* cells at 42°C resembled those of the Δtether cells at 26°C and 42°C (Fig 7A). Thus, the loss of Scs2, Scs22, and Ist2 was not additive with loss of Tcb1/2/3 at 42°C, suggesting that the Tcb proteins serve a primary role in PS homeostasis at the PM upon heat stress conditions (Fig 7A). The Ist2 protein is reportedly required for the recruitment of the Osh6 and Osh7 PS transfer proteins to ER–PM contacts (D'Ambrosio et al, 2020; Wong et al, 2021). However, loss of Ist2 had no impact on levels of the PS FLARE at the PM at either 26°C or 42°C, as compared with wild-type cells (Fig S7A). Nonetheless, we found that deletion of the *TCB1/2/3* genes in *ist2Δ* cells (generating *tcb1/2/3Δ ist2Δ* quadruple mutant cells) lowered levels of the PS FLARE at the PM at 42°C, as compared with both wild-type and *ist2Δ* cells (Fig S7A). Possibly, the Tcb proteins might serve as scaffolds for Osh6 and Osh7 ER–PM targeting under heat stress conditions. In this scenario, Ist2 may serve as an Osh6/7 tethering protein under normal growth conditions, whereas the Tcb proteins serve as Osh6/7 scaffolds during heat stress conditions. However, the Tcb proteins were not linked to Osh6/7 protein function or localisation. First of all, whereas cells lacking the Tcb proteins display significant PM integrity defects at 42°C, cells lacking Osh6 and Osh7 (*osh6/7Δ*) did not display PM integrity defects at 42°C (Fig 7B). In contrast, *cho1Δ* cells lacking PS synthase activity display PM integrity defects both at 26°C and 42°C (Fig S7B) Thus, whereas PS synthesis is required for PM integrity upon heat stress, the Osh6/7 PS transfer proteins are not. Second, the Tcb proteins are not involved in Osh7 localisation (Fig 7C). The Tcb proteins are not required for Osh7 cortical localisation at 26°C, and whereas cortical localisation of Osh7 is decreased at 42°C, there are no differences between wild type and *tcb1/2/3Δ* cells at 42°C (Fig 7C). Therefore, Osh6/7-mediated PS transport is not required to maintain PM integrity under heat stress conditions. Instead, the Tcb proteins are required for PS homeostasis and PM integrity upon heat stress conditions.

Tcb3 was shown to be specifically required for the formation of heat-induced ER–PM contacts (Collado et al, 2019). We therefore examined roles of the Tcb3 protein and its domains in PM phospholipid homeostasis and integrity upon heat-induced membrane stress. Loss of Tcb3 alone resulted in a measurable decrease in PM levels of the PS FLARE, as compared with wild-type cells at 42°C (1.5-fold; Fig 7D). This was rescued by expression of a Tcb3-GFP fusion from a plasmid, but not by a mutant form of Tcb3 lacking the SMP domain (Tcb3ΔSMP-GFP) (Fig 7D) or an artificial tether (GFP-MSP-Sac1) (Fig 7D) previously shown to restore ER–PM contacts in the Δtether cells (Manford et al, 2012). Accordingly, loss of Tcb3 alone resulted in significant PM integrity defects at 42°C (>10-fold increase; Fig 7E), consistent with a previous study (Collado et al, 2019). This was efficiently rescued by expression of Tcb3-GFP, but not by mutant forms of Tcb3 lacking either the SMP or C2 domains (Tcb3ΔSMP-GFP or Tcb3ΔC2-GFP, respectively) (Fig 7E) or by the artificial tether (GFP-MSP-Sac1) (Fig 7E).

Finally, we examined relationships between the $Ca^{2+}$-activated Tcb proteins and their roles in $Ca^2$ regulation upon heat stress conditions. Cells lacking the Tcb proteins displayed significant increases in the amplitude and duration of cytoplasmic $Ca^{2+}$ bursts at 42°C, as compared with wild-type cells (Fig 8A–C). Altogether, our findings indicate that the Tcb proteins are $Ca^{2+}$-activated membrane lipid regulatory proteins that maintain PM integrity and modulate cytoplasmic $Ca^{2+}$ signalling upon stress conditions.

## Discussion

The E-Syt/Tcb proteins have been conserved across species throughout evolution (Lee & Hong, 2006; Kopec et al, 2010; Wong & Levine, 2017), suggesting they must serve important functions. Using quantitative sensors and lipidomics, we have found that the Tcb proteins control phospholipid homeostasis at the PM. Phospholipid regulation may be an anciently conserved role of the E-Syt protein family in maintaining cellular homeostasis. In particular, localisation of a PS-specific reporter was reduced at the PM upon loss of the Tcb proteins (Figs 2B and C and 3B and C). PS has essential roles in PM organisation in eukaryotic cells (Yeung et al, 2008; Fairn et al, 2011; Cho et al, 2016; Haupt & Minc, 2017; Sartorel et al, 2018). It is intriguing that previous studies had not uncovered clear roles for the E-Syt proteins in PS regulation. However, one study found that whereas mice lacking all three E-Syt were viable, the PS transfer proteins ORP5 and ORP8 were up-regulated (Tremblay & Moss, 2016), providing a potential compensatory mechanism. Another study found no differences in PS levels in isolated PM fractions between E-Syt1/2/3 triple knockout and control cells by lipidomics (Saheki et al, 2016); however, this experiment was performed at basal cytoplasmic $Ca^{2+}$ concentrations where E-Syt1 activity is low.

---

independent experiments: all strains and conditions n = 90. ****$P$ > 0.0001; ns, not significant. **(E)** Quantitation of the percentage of wild-type and *tcb1/2/3Δ* cells with internal Pkc1-GFP puncta at 26°C and 10 min at 42°C. Data represent mean ± SD in three independent experiments (n = 50 cells per experiment). ****$P$ > 0.0001. **(F)** PM integrity assays of wild-type and *tcb1/2/3Δ* cells complemented with either an empty 2 μ vector or a 2 μ vector containing PKC1 or PKC1-T615 truncation mutant. Cells incubated at 26°C or 42°C for 10 min were subsequently incubated with propidium iodide and measured by flow cytometry (50,000 cells measured per experiment). Data represent mean ± SD from three independent experiments. ***$P$ > 0.001, ****$P$ > 0.0001; ns, not significant.
Source data are available for this figure.

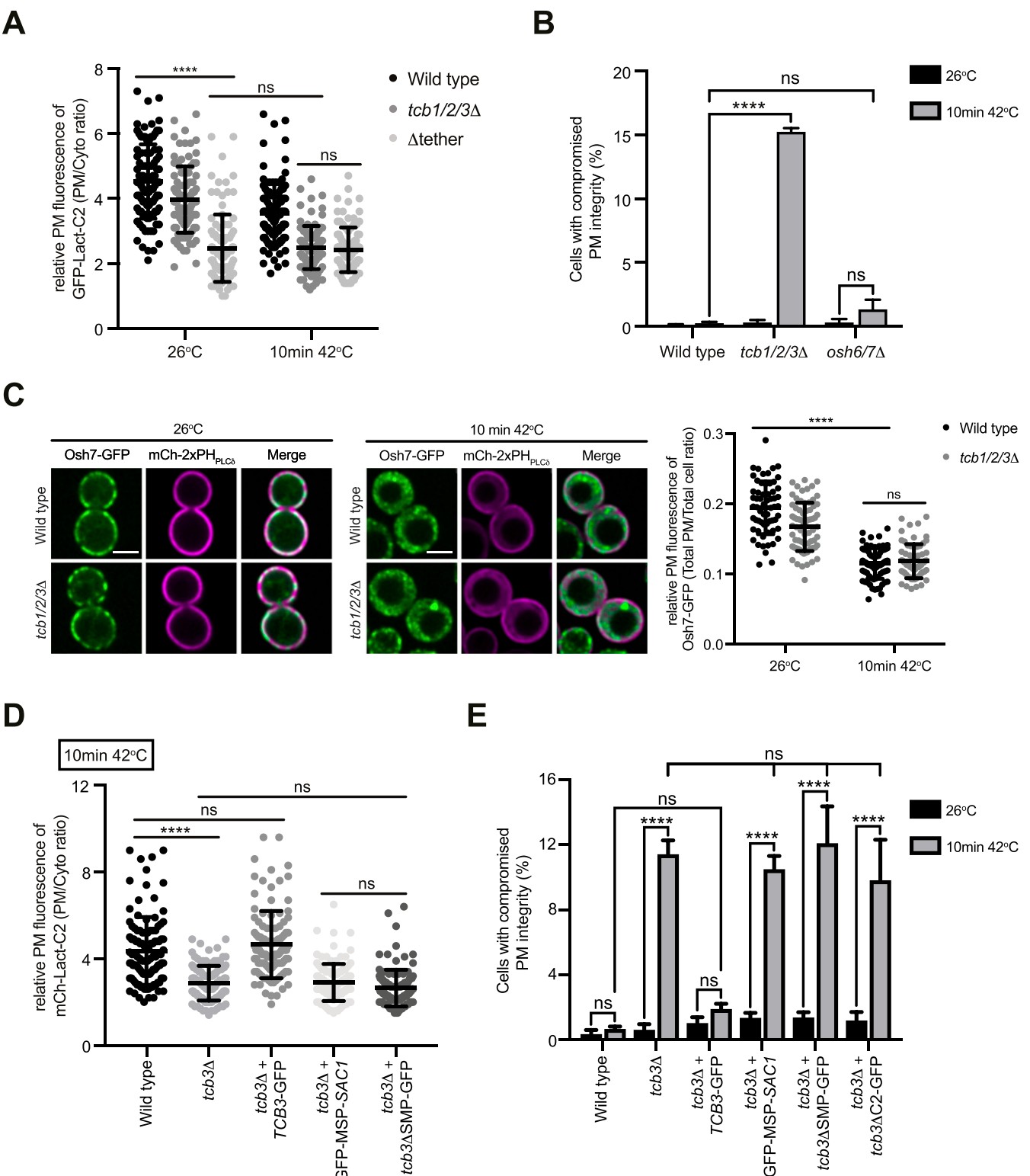

**Figure 7. Lipid transfer proteins Osh6 and Osh7 are not necessary for plasma membrane (PM) integrity, but the SMP domain of Tcb3 is required for PS distribution and PM integrity upon heat stress.**
**(A)** Quantitation of relative GFP-Lact-C2 levels at the PM in wild-type, *tcb1/2/3Δ*, and Δtether cells at 26°C or after 10 min at 42°C. Data represent mean ± SD. Total number of cells analysed in three independent experiments: all strains and conditions n = 100. ****$P > 0.0001$; ns, not significant. **(B)** PM integrity assays of wild type, *tcb1/2/3Δ* and *osh6/7Δ* cells. Cells incubated at 26°C or 42°C for 10 min were subsequently incubated with propidium iodide and measured by flow cytometry (50,000 cells measured per experiment). Data represent mean ± SD from four independent experiments. ****$P > 0.0001$; ns, not significant. **(C)** Left and middle panels: Osh7-GFP (green)

More recently, the E-Syt proteins have been implicated in $Ca^{2+}$-mediated PS externalisation (Bian et al, 2018), and this could be attributed to impaired PS delivery to the PM. Yet potential roles of the E-Syt proteins in PS regulation in mammalian cell remain to be explored.

Previous studies have reported alterations in phospholipid metabolism in Δtether yeast cells lacking several ER–PM tether proteins (Quon et al, 2018; Nishimura et al, 2019; Jorgensen et al, 2020; Wang et al, 2020). However, none of these studies addressed the specific contribution of the Tcb proteins. Two studies even questioned the roles of ER–PM contacts in lipid transfer (Quon et al, 2018; Wang et al, 2020). However, neither of these studies analysed acyl chain composition that provides important information on lipid localisation (e.g. ER versus PM species). Our species-level lipidomic results show that mono-unsaturated PS and PE, most notably 32:1 and 34:1 isoforms shown to be enriched at the PM (Schneiter et al, 1999), are specifically depleted in the Δtether cells (Fig 2E). However, there was no decrease in di-unsaturated forms of PS and PE (Fig 2E). The loss of PM specific phospholipid species is consistent with a role of ER–PM contacts in coordinating both phospholipid synthesis and transfer. All E-Syt family members feature a SMP domain that dimerizes and transports glycerolipids in vitro (Lee & Hong, 2006; Toulmay & Prinz, 2012; Schauder et al, 2014; Saheki et al, 2016; Yu et al, 2016; Bian et al, 2018; Bian & De Camilli, 2019; Qian et al, 2021). In particular, a recent study demonstrated the ability of the Tcb3 SMP domain to transport phospholipids in vitro (Qian et al, 2021), consistent with our in vivo experiments. Furthermore, the SMP domain is required for Tcb protein localisation and PM integrity (Toulmay & Prinz, 2012; Collado et al, 2019) (Fig 7E), suggesting that the SMP domain-containing Tcb proteins may directly deliver mono-unsaturated phospholipids to the PM at ER–PM contacts (Fig 9). The data do not exclude the possibility that Tcb proteins may serve as tethers for additional lipid transfer proteins at ER–PM contacts. However, the Osh6/7 PS transfer proteins were not required for PM integrity upon heat stress and the Tcb proteins were not required for Osh7 cortical localisation (Figs 7 and 9).

E-Syt family members have been implicated in distinct steps of the phosphoinositide cycle in metazoan cells. For example, E-Syt proteins have been suggested to facilitate the activity of certain phosphatidylinositol transfer proteins (PITPs) at ER–PM contacts, including Nir2/3 and Rgdβ in mammalian cells and *Drosophila*, respectively (Chang et al, 2013; Nath et al, 2020). The Nir2/3 and Rgdβ PITPs are proposed to transfer PI from the ER to the PM for the generation of PI4P and $PI(4,5)P_2$ at the PM. However, PI4P synthesis at the PM is not impaired upon loss of the Tcb proteins or other ER–PM tethers, Scs2/22 and Ist2 (Figs 1D, E, and G and S5C) (Manford et al, 2012). Thus, the Tcb proteins and other ER–PM contact proteins (Scs2/22 and Ist2) are not required for delivery of PI to the PM in yeast. Furthermore, orthologs of the Nir2/Rgdβ PITPs do not exist in fungi (Hsuan & Cockcroft, 2001), and so the putative role of E-Syt proteins in PI transfer may have evolved separately in animals. Mammalian cells lacking the E-Syt proteins also feature prolonged accumulation of DAG at the PM after phospholipase C (PLC)-mediated $PI(4,5)P_2$ hydrolysis (Saheki et al, 2016), suggesting a defect in recycling DAG to the ER during the phosphoinositide cycle. We did not observe accumulation of a bona fide sensor of Plc1-generated DAG at the PM or changes in DAG levels in cells lacking the Tcb proteins (Figs 5F and S5A). However, because DAG can undergo transbilayer movements between the cytoplasmic and extracellular membrane leaflets, these results do not rule out a potential role of the Tcbs, or other SMP domain-containing proteins, in channelling DAG at intra-organelle contacts.

ER-localised Tcb3 is found in proximity to the PM-localised Sfk1 protein, and co-localisation of Tcb3 with Sfk1 increases after heat stress (Fig 4). Sfk1 (TMEM150 ortholog) is reportedly a subunit of a PI 4-kinase (PI4K) complex that generates PI4P at the PM and is required for heat-stimulated $PI(4,5)P_2$ synthesis (Audhya & Emr, 2002). Steady-state lipidomics, metabolic labelling, and quantitative microscopy experiments reveal that Sfk1 and the Tcb proteins are not required for PI4P and $PI(4,5)P_2$ synthesis (Figs 1 and S5) (Omnus et al, 2020). However, PI4P 5-kinases are recruited and activated by PS (Fairn et al, 2009; Nishimura et al, 2019), and regulation of PS by Sfk1 and the Tcb proteins may indirectly regulate $PI(4,5)P_2$ synthesis at the PM. Interestingly, Sfk1 is also implicated in PM phospholipid asymmetry and PM integrity (Mioka et al, 2018; Kishimoto et al, 2021). As $PI(4,5)P_2$ synthesis is required for PM integrity (Audhya et al, 2000; Omnus et al, 2016), one possibility is that Tcb3 and Sfk1 promote the formation of PS-enriched domains that promote PI4P 5-kinase activity upon PM stress. In support of this, the Tcb proteins are required for PS homeostasis at the PM upon heat stress (Figs 3 and 7D). Accordingly, the Tcb proteins are activated upon heat-induced $Ca^{2+}$ signalling and form distinct ER–PM contacts that are required for PM integrity under these conditions (Fig 7D) (Omnus et al, 2016; Collado et al, 2019).

We also show that the Tcb proteins facilitate heat-induced recruitment of Pkc1 to the mother cell cortex that is necessary for PM integrity upon heat stress (Fig 6). Pkc1 contains a C2 domain that controls Pkc1 localisation at the PM, and PS has been shown to be required for Pkc1 function (Denis & Cyert, 2005; Dey et al, 2017; Nomura et al, 2017). Accordingly, mammalian non-conventional PKC family members are regulated by PS (Heinisch & Rodicio, 2018).

localisation in wild-type and *tcb1/2/3Δ* mutant cells, co-expressing the PM marker mCh-2xPH$_{PLCδ}$ (magenta), at 26°C (left) and following a brief heat stress (10 min at 42°C) (middle). Scale bar, 4 μm. Right panel: quantitation of relative Osh7-GFP levels at the PM in wild-type and *tcb1/2/3Δ* cells at 26°C or after 10 min at 42°C. Data represent mean ± SD. Total number of cells analysed in four independent experiments: wild type 26°C n = 61, wild type 10 min 42°C n = 76, *tcb1/2/3Δ* 26°C n = 71, *tcb1/2/3Δ* 10 min 42°C n = 76 cells. ****$P > 0.0001$; ns, not significant. **(D)** Quantitation of relative GFP-Lact-C2 levels at the PM in wild type, *tcb3Δ*, and *tcb3Δ* cells expressing *TCB3*-GFP, a mutant Tcb3 fusion lacking the SMP domain (*tcb3ΔSMP-GFP*), or an artificial ER–PM tether (GFP-MSP-*SAC1*) (Manford et al, 2012) after 10 min at 42°C. Data represent mean ± SD. Total number of cells analysed in three independent experiments: wild type n = 108, *tcb3Δ* n = 116, *tcb3Δ* + *TCB3*-GFP n = 108, *tcb3Δ* + *tcb3ΔSMP-GFP* n = 112, *tcb3Δ* + GFP-MSP-*SAC1* n = 112. ****$P > 0.0001$; ns, not significant. **(E)** PM integrity assays of wild-type, *tcb3Δ*, and *tcb3Δ* cells complemented with *TCB3*-GFP, *TCB3*-GFP truncation mutants (*tcb3ΔSMP-GFP* or *tcb3ΔC2-GFP*), or GFP-MSP-*SAC1*. Cells incubated at 26°C or 42°C for 10 min were subsequently incubated with propidium iodide and measured by flow cytometry (50,000 cells measured per experiment). Data represent mean ± SD from three independent experiments. ****$P > 0.0001$; ns, not significant. Also see Fig S7.
Source data are available for this figure.

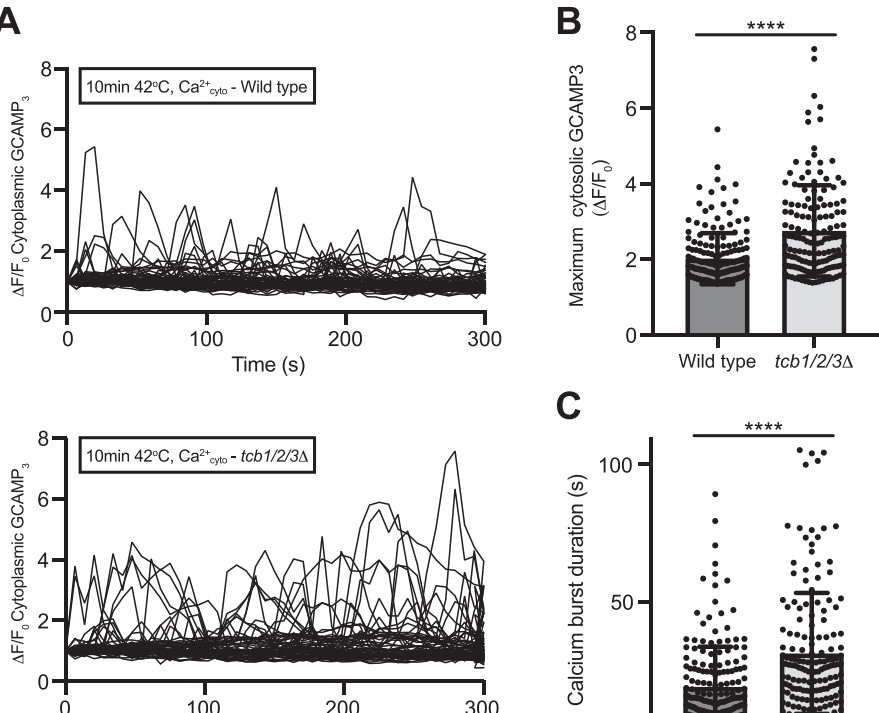

**Figure 8.  The tricalbins regulate cytosolic $Ca^{2+}$ burst intensity upon heat stress.**
**(A)** Normalised specific GCaMP3 fluorescence ($\Delta F/F_0$) traces of wild type and *tcb1/2/3Δ* cells incubated at 42°C for 10 min. Each line represents an individual cell, 62 (top) and 59 (bottom) traces for each condition.
**(B, C)** Mean maximum normalised specific fluorescence ($\Delta F/F_0$) (B) and duration of calcium bursts (C) in wild-type and *tcb1/2/3Δ* cells 5 min after incubation at the temperature indicated. Data represent mean ± SD. Total number of cells analysed in three independent experiments: wild type 10 min 42°C n = 150, *tcb1/2/3Δ* 10 min 42°C n = 148. ****$P > 0.0001$.
Source data are available for this figure.

Correct maintenance of phospholipids within the PM may be essential for Pkc1 PM recruitment. Consistent with this, loss of Sfk1 also impaired Pkc1 cortical assembly upon heat stress (Fig S6). However, our results do not rule out roles of Sfk1 and the Tcb protein in regulating other lipids necessary for PM maintenance and integrity. For example, the Tcb proteins have recently been implicated in non-vesicular transport of ceramide from the ER to Golgi compartments (Ikeda et al, 2020). It will be interesting to examine whether the Tcb proteins transfer sphingolipids from the ER to the PM upon heat stress.

Correct maintenance of PS within the PM is integral to cell survival (Matsuo et al, 2007) and in mammalian cells, rearrangements in PS distribution within the PM bilayer acts as a trigger for apoptotic cell death (Fadok et al, 1992). PS also accumulates at sites of PM damage (Horn & Jaiswal, 2019) and is involved in the recruitment of $Ca^{2+}$- and PS-binding proteins such as annexins, dysferlins, and synaptotagmins during PM repair (Middel et al, 2016; Boye et al, 2017; Horn & Jaiswal, 2018). However, the mechanism for PS enrichment at sites of PM damage is incompletely understood. Regulation of PS dynamics may be an anciently conserved function of E-Syt family members in PM integrity and repair. The synaptotagmin proteins have been the subject of intense focus because of their functions in neurotransmitter release at neuronal synapses (Südhof, 2002; Chapman, 2008), although they also act as $Ca^{2+}$-dependent regulators of exocytic events in non-neuronal exocrine cells, driving SNARE-mediated membrane fusion (Hui et al, 2009; van den Bogaart et al, 2011; Kim et al, 2012). Accordingly, synaptotagmins and dysferlins are proposed to aid in PM repair by triggering vesicle fusion at sites of injury (Detrait et al, 2000; Lek et

al, 2013). Synaptotagmin family members are not conserved in fungi, unlike the extended-synaptotagmins (E-Syts) that are expressed ubiquitously throughout the eukaryotic as well as prokaryotic kingdoms (Wong & Levine, 2017). This hints to a more ancient function of the E-Syt proteins that predates that of the C2 domain-containing synaptotagmins. Protein kinase C isoforms that bear a C2 domain may be a common ancestral effector protein involved in cellular integrity and exocytosis in yeast and mammalian cells (Levin, 2005; Horn and Jaiswal, 2018, 2019). Therefore, the E-Syt family may confer a primordial mechanism for PM maintenance that subsequently evolved in higher eukaryotes in the control of regulated exocytosis during PM wound repair and eventually even insulin and neurotransmitter release.

The importance of $Ca^{2+}$ signalling in membrane stress responses cannot be understated. Weakening or disruption of the PM can lead to influx of $Ca^{2+}$ into the cytoplasm, resulting in an increase in cytoplasmic $Ca^{2+}$ concentration up to two orders of magnitude (from 0.1 $\mu$M to up to 10 $\mu$M) (Jaiswal, 2001). Although $Ca^{2+}$ is recognised as a signal for membrane damage response and repair pathways, increases in cytoplasmic $Ca^{2+}$ above a defined threshold (>10 $\mu$M) can result in loss of membrane organelle and cellular integrity leading to cell death, as originally described by Zimmerman and Hülsmann as "the calcium paradox" (Zimmerman & Hülsmann, 1966). Accordingly, cells have established rapid response systems to damage- and stress-induced $Ca^{2+}$ influx. Many of the proteins involved in PM repair mechanisms (including exocytosis-mediated vesicle fusion, endocytosis, membrane shedding, and reorganisation of the cytoskeleton in response to PM stress or injury) are $Ca^{2+}$-binding proteins that function as $Ca^{2+}$

**Figure 9. Speculative model for tricalbin protein function in plasma membrane (PM) phospholipid homeostasis and integrity.**
Under normal growth conditions (left panels), non-vesicular phosphatidylserine (PS, magenta) transport from the ER to the PM is carried out by Osh6 and Osh7 that are recruited to ER–PM contacts by Ist2 (Maeda et al, 2013; Moser von Filseck et al, 2015; D'Ambrosio et al, 2020; Wong et al, 2021). Osh6 and Osh7 move PS from the ER to PM in exchange for PI4P (yellow) at the PM (top left, wild-type cells). Under normal growth conditions, the tricalbin (Tcb) proteins may have a structural role in tethering the ER and PM, but they are not required for PS transport as loss of Tcb1/2/3 does not substantially affect PS distribution (bottom left, *tcb1/2/3Δ* cells). Under heat stress conditions, however, increases in cytoplasmic Ca$^{2+}$ levels further induce Tcb3 function. This may include dimerisation with Tcb1/2 and interactions with the PM through the C2 domains that may promote the ability of the Tcbs to form heat-induced ER–PM contacts (Collado et al, 2019). Importantly, under heat stress conditions, the Tcbs serve a primary role in PS homeostasis at the PM, either by SMP domain-mediated non-vesicular PS transfer from the ER to the PM (top right) or by recruiting additional PS transfer proteins to ER–PM contacts. Tcb proteins also undergo a heat-dependent association with PM-localised Sfk1 that has been implicated in regulating phospholipid asymmetry out at the PM. Loss of the Tcb proteins (bottom right, *tcb1/2/3Δ*) leads to a lack of PS delivery to the PM in heat stressed cells, resulting in loss of Pkc1 recruitment (see Fig 5), as well as prolonged Ca$^{2+}$ bursts and PM integrity defects (bottom right, *tcb1/2/3Δ*).

sensors (Benink & Bement, 2005; Cheng et al, 2014; Holmes et al, 2015; Horn & Jaiswal, 2018; Koerdt et al, 2019). The E-Syt family members are no exception and function as sensors of large increases in cytoplasmic Ca$^{2+}$ (>1 μM) (Idevall-Hagren et al, 2015; Ge et al, 2021). Accordingly, mammalian E-Syt proteins are regulated by the store-operated Ca$^{2+}$ entry (SOCE) pathway and subsequently inactivate SOCE (Giordano et al, 2013; Idevall-Hagren et al, 2015). We find that the yeast Tcbs are active under conditions that increase cytoplasmic Ca$^{2+}$ signals (Figs 1 and S1) and the Tcbs in turn attenuate cytoplasmic Ca$^{2+}$ signalling (Fig 8). Thus Tcb/E-Syt function at ER–PM contacts responds to and modulates cytoplasmic Ca$^{2+}$ signals, and this regulatory system is conserved from yeast to humans.

In summary, we propose that the yeast Tcb proteins serve as Ca$^{2+}$-activated ER–PM tethers and lipid transfer proteins that maintain PS levels necessary for PM integrity under stress conditions (Fig 9). Tcb-mediated regulation of PS in the cytosolic leaflet of the PM may aid in the direct recruitment of several protein and lipid

kinases required for PM integrity. Essential roles of the mammalian E-Syt proteins in cell and tissue homeostasis may be revealed under conditions of PM stress conditions, and it will be interesting to re-examine the E-Syt proteins in this context.

# Materials and Methods

### Yeast strains, plasmids, media, and growth assays

Descriptions of strains and plasmids used in this study used in this study are listed in Tables S5 and S6. Gene deletions and epitope tags were introduced into yeast by homologous recombination (Longtine et al, 1998). The pRS vector series have been described previously (Sikorski & Hieter, 1989). Plasmids were sequenced to ensure that no mutations were introduced because of manipulations. Standard techniques and media were used for yeast and

bacterial growth. For plating assays in Fig S1, cells were grown to midlog, adjusted to 1 OD600/ml, and serial dilutions were plated on agar media either containing or lacking 0.75 $\mu$M myriocin. Haploid Split-GFP strains listed in Table S5 were made using tagging cassettes (Barnard et al, 2008). Diploid strains were then created by mating to obtain various combinations of N- and C-terminal GFP fragment-tagged target proteins.

## Live yeast cell imaging

Fluorescence microscopy experiments were performed on mid-log yeast cultures in synthetic media at the indicated temperatures. Live yeast cell imaging data in all Figures were acquired with a 100× CFI Plan Apochromat VC oil-immersion objective lens (1.4 NA), using a PerkinElmer Ultraview Vox spinning disk confocal microscope that consists of a Nikon TiE inverted stand attached to a Yokogawa CSU-X1 spinning disk scan head, a Hamamatsu C9100-13 EMCCD camera, Prior NanoscanZ piezo focus, and a Nikon Perfect Focus System (PFS). All images were collected as square images with 512 × 512 pixels. The number of cells observed in experiments is reported in the figures and figure legends. The brightness and contrast of images were linearly adjusted and cropped in Photoshop (Adobe) for presentation.

For vacuole staining in Fig S3, 5 $\mu$M FM4-64 (Invitrogen) was added to mid-log cell cultures in YPD media for 15 min at 26°C. Cultures were then resuspended in fresh YND media and incubated for 1 h in at 26°C. 0.1 mM CellTracker Blue CMAC (Invitrogen) was then added 15 min before heat stress and imaging. For the myriocin treatment in Fig S6, 2 $\mu$M myriocin was added to mid-log cell cultures in YND media for 1 h at 26°C before heat stress and imaging.

## Quantitative image analysis

All quantitative image analyses were conducted using ImageJ/Fiji (Schindelin et al, 2012). To calculate the specific PM to cytosol (PM/Cyto) ratio of individual lipid species in Figs 1 and 2, S2, 3, 4, S5, 7, and S7, the relative fluorescence (relative $F_{PM}$) was quantified as described in Nishimura et al (2019). Briefly, individual cells were chosen from single channel images, lines were drawn cross the mother cell, and the corresponding fluorescence intensity profiles were plotted. The two highest intensity values, corresponding to signal at the PM, were averaged ($F_{PM}$). Intensity measurements were also taken from lines drawn through the cytosol ($F_{cytosol}$) and background ($F_{background}$) and PM relative fluorescence was calculated by using the equation: relative $F_{PM} = (F_{PM} - F_{background})/(F_{cytosol} - F_{background})$. Peaks in intensity profiles were automatically calculated by an Excel VBA macro.

Line profiles to visualise colocalisation of GFP-Lact-C2 and vacuole or nER markers in Figs 3 and S3 were generated from intensity measurements taken along a straight line drawn through the whole cell. To calculate the nER to cytosol (nER/cytosol) and vacuole membrane to cytosol (vac/cytosol) ratios of GFP-Lact-C2 in Figs 3 and S3, individual lines were drawn that pass twice through either the nER or vacuole membranes. The corresponding fluorescence intensity profiles were plotted and the two highest GFP-Lact-C2 intensity values, which coincided with the respective

membrane marker (DsRed-HDEL or FM4-64 staining) were averaged ($F_{nER}$ or $F_{vac}$) and used to calculate the relative fluorescence (relative $F_{nER}$ or relative $F_{vac}$) as described above. To calculate the relative Pkc1-GFP intensity at the PM of the mother cell or daughter bud in Figs 6 and S6, lines were manually drawn along the PM of the mother and daughter cells, as identified using the PM marker, mCherry-2xPH$_{PLC\delta}$. The average intensity along each line ($F_{PM}$) was determined and the PM relative fluorescence (relative $F_{PM}$) was individually calculated for the mother and daughter cells as previously described. To identify the percentage of cells with distinct internal Pkc1-GFP puncta in Fig 6, maximum projections of individual cells were selected and for each cell, points of interest (Pkc1-GFP foci) were identified using the Find Maxima tool in Fiji, applying appropriate noise tolerance settings. Split GFP (Barnard et al, 2008) signal intensity in Figs 4 and S4 was measured using Fiji.

## LC–MS/MS analysis of methylated PIPs

Levels of individual PI, PIP and PIP2 species in wild type, *tcb1/2/3Δ*, *scs2/22Δ ist2Δ* and Δtether mutant cell extracts Figs 1 and S1 were analysed as previously described (Nishimura et al, 2019). 20 OD$_{600}$ units of cells were precipitated and washed with cold 4.5% perchloric acid. For phosphoinositide measurements, cells were resuspended in 500 $\mu$l 0.5 M HCl and disrupted with a 5.0 mm zirconia bead by vigorous shaking (1,500 rpm for 10 min, Shake Master Neo [BMS]). The homogenates were transferred to new tubes and centrifuged at 15,000$g$ for 5 min. The pellets were resuspended in 170 $\mu$l water and 750 $\mu$l of CHCl$_3$/MeOH/1 M HCl (2:1: 0.1, vol/vol) and incubated for 5 min at room temperature. To each sample, 725 $\mu$l of CHCl$_3$ and 170 $\mu$l of 2 M HCl were added, followed by vortexing. After centrifugation at 1,500$g$ for 5 min, the lower phase was collected and washed with 780 $\mu$l of pre-derivatization wash solution (the upper phase of CHCl$_3$/MeOH/0.01 M HCl [2:1:0.75 vol/ vol]). The lipid extracts were derivatized by adding 50 $\mu$l of 2 M TMS-diazomethane in hexane. The derivatization was carried out at room temperature for 10 min and was stopped by adding 6 $\mu$l of glacial acetic acid. The derivatized samples were washed twice with 700 $\mu$l of post-derivatization wash solution (the upper phase of CHCl$_3$/ MeOH/water [2:1:0.75 vol/vol]). After adding 100 $\mu$l of MeOH/H$_2$O (9: 1, vol/vol), the samples were dried under a stream of N$_2$, dissolved in 80 $\mu$l of MeOH and sonicated briefly. After adding 20 $\mu$l of water, the samples were subjected to LC-ESI-MS/MS analysis. The LC-ESI-MS/MS analysis was performed on a Shimadzu Nexera ultra high-performance liquid chromatography system coupled with a QTRAP 4500 hybrid triple quadrupole linear ion trap mass spectrometer. Chromatographic separation was performed on an Acquity UPLC C4 BEH column (100 × 1 mm, 1.7 $\mu$m; Waters) maintained at 40°C using mobile phase A (water containing 0.1% formate) and mobile phase B (acetonitrile containing 0.1% formate) in a gradient program (0–5 min: 45% B; 5–10 min: 45% B→100% B; 10–15 min: 100% B; 15–16 min: 100% B→45% B; 16–20: 45% B) with a flow rate of 0.1 ml/min. The instrument parameters for positive ion mode were as follows: curtain gas, 10 $\psi$; collision gas, 7 arb. unit; ionspray voltage, 4,500 V; temperature, 600°C; ion source gas 1, 30 $\psi$; ion source gas 2, 50 $\psi$; declustering potential, 121 V; entrance potential, 10 V; collision energy, 39 V; collision cell exit potential, 10 V. Methylated phos-phoinositides and phosphatidylserine were identified and

quantified by multiple reaction monitoring. For these measurements, internal standards of 10 ng of 17:0–20:4 PI, PI(4)P, and PI(4,5)$P_2$, and PS were added to each sample.

## Quantitative shotgun lipid MS data acquisition, analysis, and post-processing

Levels of individual PS, PE, PC, PA, PI, and DAG species shown in Figs 2, S2, and 5, were determined by mass spectrometry-based quantitative, shotgun lipidomics by Lipotype GmbH as described (Ejsing et al, 2009; Klose et al, 2012). Total yeast cell lysate samples were diluted to 0.2 OD units using 155 mM ammonium bicarbonate in water to the total volume of 150 $\mu$l and were spiked with internal lipid standard mixture. Lipids were extracted using a two-step chloroform/methanol procedure with 750 $\mu$l volume of each organic phase step (chloroform:methanol, 15:1 and 2:1, respectively for the first and the second step) (Ejsing et al, 2009). After extraction, the organic phase was transferred to an infusion plate and dried in a speed vacuum concentrator. First step—the dry extract was resuspended in 100 $\mu$l 7.5 mM ammonium acetate in chloroform/methanol/propanol (1:2:4, V:V:V); second step—the dry extract in 100 $\mu$l 33% ethanol solution of methylamine in chloroform/methanol (0.003:5:1; V:V:V). All liquid handling steps were performed using Hamilton Robotics STARlet robotic platform with the Anti Droplet Control feature for organic solvents pipetting. Samples were analysed by direct infusion on a QExactive mass spectrometer (Thermo Fisher Scientific) equipped with a TriVersa NanoMate ion source (Advion Biosciences). Samples were analysed in both positive and negative ion modes with a resolution of $R_{m/z=200}$ = 280,000 for MS and $R_{m/z=200}$ = 17,500 for MSMS experiments, in a single acquisition. MSMS was triggered by an inclusion list encompassing corresponding MS mass ranges scanned in 1 D increments (Surma et al, 2015). Both MS and MSMS data were combined to monitor EE, DAG and TAG ions as ammonium adducts; PC as an acetate adduct; and PA, PE, PG, PI, and PS as deprotonated anions. Data were analysed with in-house developed lipid identification software based on LipidXplorer (Herzog et al, 2011, 2012). Data post-processing and normalization were performed using an in-house developed data management system. Only lipid identifications with a signal-to-noise ratio >5 and a signal intensity fivefold higher than in corresponding blank samples were considered for further data analysis.

## Analysis of $^3$H-labeled inositol phosphates by HPLC

PIP levels were analysed as previously described (Nishimura et al, 2019). Briefly, 5 OD600 units of cells cultured in YND medium were washed by medium lacking inositol and pre-incubated at 26°C or 42°C for 15 min. The cells were labeled with 50 $\mu$Ci of myo-[2-H3]-inositol in medium lacking inositol and further incubated for 1 h. Then, the cells were lysed in 4.5% perchloric acid with glass beads to generate extracts. After washed by 0.1 M EDTA, the extracts were mixed with methylamine reagent (methanol/40% methylamine/water/1-butanol; 4.6:2.6:1.6:1.1 vol/vol) and incubated at 53°C for 1 h to deacylate phospholipids. Samples were dried in a vacuum chamber, washed with water, dried again, and resuspended in 300 $\mu$l water. Extraction reagent (1-butanol/ethl-ether/formic acid ethyl ester; 20:4:1 vol/vol) was added and [3H] glycerol-PIPs were separated into the aqueous phase by vortexing and centrifugation at 14,000$g$ for 5 min. The extraction was repeated twice and the final aqueous phase was collected and dried. Dried pellets were resuspended in 260 $\mu$l water and separated on a PartiSphere 5-$\mu$m SAX column attached to a PerkinElmer Series 200 HPLC system and a radiomatic 150TR detector using Ultima-Flo AP scintillation fluid. The HPLC and on-line detector were controlled with Total Chrome Navigator software. The data were analysed using Total Chrome Navigator software.

## Quantitative GCaMP3 fluorescence assays

To quantify resting cytosolic $Ca^{2+}$ levels in Figs 1 and S1, strains expressing a cytoplasmic GCaMP3 reporter were grown at 26°C. For flow cytometry experiments, cells were transferred to PBS. Mean fluorescence of 50,000 events was recorded on a BD Accuri C6 flow cytometer. Background was determined using strains harbouring vector alone. For fluorescence microscopy measurements cells were transferred immediately to a slide and imaged directly. For identification of $Ca^{2+}$ bursts after heat stress in Figs 3, S3, and 8, cells were incubated for 10 min at the indicated temperature before being directly transferred to a slide and imaged at max speed for 5 min. Individual cells were chosen from single channel time lapse images. Intensity measurements were taken in an area of the same size within the cell ($F_{cell}$) and adjacent background ($F_{background}$). The normalised GCaMP3 signal ($\Delta F/F_0$) was calculated from time lapse images, using the equation: $\Delta F/F_0 = (F_{cell}$ at time $t - F_{background}$ at time $t)/(F_{cell}$ at time $t = 0 - F_{background}$ at time $t = 0$). Peaks in GCaMP3 fluorescence intensity, representing $Ca^{2+}$ bursts, were identified using Prism 8 Area under curve function.

## PM integrity assay

PM integrity assays in Figs 6 and 7 and S7 were performed as described (Omnus et al, 2016; Collado et al, 2019). Yeast strains were grown at 26°C to midlog phase, then kept at 26°C or shifted to either 42°C for 10 min. 1 $OD_{600}$ equivalent of cells was pelleted, resuspended in PBST (0.01% Tween 20), and stained with propidium iodide (Sigma) for 15 min. Cells were then washed twice with PBST and analysed by flow cytometry on a BD Accuri C6 flow cytometer. For flow cytometry analysis, 50,000 cells were counted for each sample from three independent experiments and combined for a total analysis of 150,000 cells. Background was determined by analysing each of the cell strains at the indicated temperatures before staining with propidium iodide.

## Quantification and statistical analysis

Statistical analysis was carried out using GraphPad Prism 8. To compare the mean of two groups, an unpaired two-tailed $t$ test was used. To compare the mean of multiple groups, we used one-way ANOVA followed by Tukey–Kramer multiple comparisons.

# Data Availability

Materials used in this study are available from the corresponding author on request.

# Supplementary Information

# Acknowledgements

We thank Scott Emr, Yuxin Mao, Martha Cyert, Sophie Martin, and Greg Fairn for strains and plasmids. We also thank Ruben Fernandez-Busnadiego, Tim Levine, Taki Nishimura, and Bailey Hewlett for helpful discussions. CJ Stefan is supported by Medical Research Council funding to the Medical Research Council LMCB University Unit at University College London, award code MC_UU_00012/6.

## Author Contributions

FB Thomas: conceptualization, data curation, formal analysis, validation, investigation, visualization, methodology, and writing—original draft, review, and editing.
DJ Omnus: conceptualization, investigation, visualization, and methodology.
JM Bader: conceptualization, investigation, visualization, and methodology.
GHC Chung: conceptualization, investigation, visualization, and methodology.
N Kono: conceptualization, data curation, validation, investigation, visualization, and methodology.
CJ Stefan: conceptualization, resources, supervision, funding acquisition, project administration, and writing—original draft, review, and editing.

## Conflict of Interest Statement

The authors declare that they have no conflict of interest.

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
