## [Reviewer comments · Life Science Alliance]

Life Science Alliance

Tricalbin proteins regulate plasma membrane phospholipid homeostasis

Ffion Thomas, Deike Omnus, Jakob Bader, Gary Chung, Nozomu Kono, and Christopher Stefan

DOI: <https://doi.org/10.26508/lsa.202201430>

Corresponding author(s): Christopher Stefan, University College London

Review Timeline:

Submission Date:	2022-02-28
Editorial Decision:	2022-03-21
Revision Received:	2022-04-03
Accepted:	2022-04-04

Transaction Report:

Please note that the manuscript was reviewed at Review Commons and these reports were taken into account in the decision-making process at Life Science Alliance.

March 21, 2022

RE: Life Science Alliance Manuscript #LSA-2022-01430

Dr. Christopher J Stefan
University College London
MRC Laboratory for Molecular Cell Biology
Gower Street
London, Greater London WC1E 6BT
United Kingdom

Dear Dr. Stefan,

Thank you for submitting your revised manuscript entitled "Tricalbin proteins regulate plasma membrane phospholipid homeostasis". We would be happy to publish your paper in Life Science Alliance pending final revisions necessary to meet our formatting guidelines.

- please respond to Reviewer 1's comments about the claim suggesting Tcbs directly transport PS. No additional experimentation is expected.
- please upload your main manuscript text as an editable doc file
- please upload your main and supplementary figures as single files
- please add a Running Title and a Summary Blurb/Alternate Abstract to our system
- please add a Category for your manuscript in our system
- please add the Twitter handle of your host institute/organization as well as your own or/and one of the authors in our system
- please add the Authors' Contributions to the system as well
- please add a callout for Figure 5F to your main manuscript text

A. FINAL FILES:

B. MANUSCRIPT ORGANIZATION AND FORMATTING:

Sincerely,

Reviewer #1 (Comments to the Authors (Required)):

This study has been improved, but my broader concerns remain the same.

The primary conceptual advance is the proposal that the Tcbs regulate PS (and perhaps PE) levels at the PM. The findings are consistent with this, but there is no mechanistic insight, significantly limiting the impact of the work. The authors suggest Tcbs directly transport PS (and perhaps other lipids), but this claim can only be strongly supported by direct measurement of lipid transport rates in cells. Changes in steady levels of lipids or the distribution of lipid sensors do not, by themselves, indicate that lipid transport is altered. It is better to directly assess transport. This is challenging, but has been done before; for example, in papers assessing the role of Osh6 or Osh7 in PS transport. If the Tcbs do not mediate PS transport, some mechanistic insight into how they affect PS distribution would significantly increase the impact of this study.

Similarly, the study now makes a stronger case that the Tcbs are functionally linked to Pkh1 and Skh1, but there is still no mechanistic insight into how this occurs.

In summary, there is a lot of high-quality data here and proposing that the Tcbs transport PS is certainly reasonable, but the idea that Tcbs directly transport PS or are involved in PS distribution is not terribly convincing.

Reviewer #2 (Comments to the Authors (Required)):

The authors have added a considerable amount of additional evidence to show the role of Tricalbins and their role in PS transport. The work is carefully done with a number of controls. The work brings fresh insight into the role of Tricalbins.

Response to reviewers:

We thank the each of the reviewers for evaluating our revised manuscript.

Reviewer #1 (Comments to the Authors (Required)):

This study has been improved, but my broader concerns remain the same. The primary conceptual advance is the proposal that the Tcb3 regulate PS (and perhaps PE) levels at the PM. The findings are consistent with this, but there is no mechanistic insight, significantly limiting the impact of the work.

Response: The quantitative data in our study definitively show that the tricalbin (Tcb) proteins regulate phosphatidylserine (PS) metabolism and distribution. We also show that the Tcb3 SMP domain (a proposed lipid transfer domain) is required for maintenance of PS at the plasma membrane (PM) upon heat stress. However, we acknowledge that these findings lack 'mechanistic' detail, especially at the molecular level. To address the reviewer's criticism that our study lacks 'mechanistic' insight, we have removed the word "mechanistic" from the manuscript.

We disagree that our study lacks 'impact'. While the Tcb3 SMP domain has been shown to directly transfer phospholipids *in vitro* (Qian et al., 2021), our study is the first to show that the Tcb3 protein regulates PS distribution *in vivo*. As mentioned, we show that deletion of the Tcb3 SMP domain results in PS distribution defects upon heat stress. There are also other novel findings in our study that are highly impactful. First, our study provides new insight into the roles of the Tcb protein in PM integrity. In particular, new results in the revised manuscript demonstrate that the Tcb proteins are required for efficient recruitment of Pkc1, a known PS-binding protein that is essential for cellular integrity, to the mother cell cortex upon heat stress conditions. In addition, our study is first to demonstrate that Tcb3 and Sfk1 co-localise and are functionally linked in the control of PM lipid organisation.

The authors suggest Tcb3 directly transport PS (and perhaps other lipids), but this claim can only be strongly supported by direct measurement of lipid transport rates in cells. Changes in steady levels of lipids or the distribution of lipid sensors do not, by themselves, indicate that lipid transport is altered. It is better to directly assess transport. This is challenging, but has been done before; for example, in papers assessing the role of Osh6 or Osh7 in PS transport.

Response: Our quantitative lipidomics and microscopy results together (not alone) strongly indicate that the Tcb proteins regulate PS distribution *in vivo*. Strong evidence is also provided by numerous studies showing that SMP domains transport phospholipids *in vitro* (Schauder et al., 2014; Saheki et al., 2016; Yu et al., 2016; Bian et al., 2018; Bian and De Camilli, 2019; Qian et al., 2021). In particular, a recent study has demonstrated that the Tcb3 SMP domain directly transports phospholipids *in vitro* (Qian et al., 2021). Based on the existing *in vitro* results and our new *in vivo* findings, we suggest that the Tcb proteins may serve as PS transfer proteins *in vivo*, but we also discuss alternative possibilities in the manuscript (e.g. in the Discussion and the Figure 9 legend). We have not overstated the results in the study.

We agree that cellular lipid transport assays are valuable and informative. However, they do have limitations. Despite the reviewer's claim, *in vivo* lipid transfer assays cannot demonstrate that the Tcb proteins "directly transport" PS. They do not discern whether a protein directly transports a lipid or instead indirectly facilitates the function of another protein that actually carries out the lipid transfer reaction.

Even so, we agree that cellular PS transport assays, such as those used to monitor Osh6- and Osh7-mediated PS transport (Maeda et al., 2013; Moser von Filseck et al., 2015; D'Ambrosio et al., 2019), are useful. However, they are also somewhat challenging to perform, as the reviewer pointed out. Unfortunately, adapting this assay for our purposes presented some additional experimental challenges. Firstly, it requires the generation of Δ tether cells in which the *CHO1* gene is also deleted, resulting in cells that are very sick. In addition, *tcb3* single mutant cells only display PS reporter phenotypes upon heat stress conditions. However, it has been observed that heat stress causes increased endocytic internalisation of NBD-labelled phospholipids, making it difficult to distinguish between vesicular and non-vesicular lipid trafficking. Disruptions in endocytosis have been shown to modulate non-vesicular lipid transport pathways, making it difficult to circumvent this problem. We have yet to establish experimental conditions to strictly monitor non-vesicular lipid trafficking at high temperature and further work is needed to establish a reliable assay. For these reasons, we have had to rely on results from quantitative steady-state lipidomics and microscopy experiments to base our conclusions.

In a future study, we will aim to develop cellular assays to monitor the role of ER-PM contacts and the Tcb proteins in PS transport. We also recognise that these experiments may be informative in monitoring many transport steps including incorporation of lyso-PS into the PM, flip/flop of lyso-PS at the PM, retrograde transport of lyso-PS to the ER, conversion of lyso-PS to PS, and anterograde transport of PS to the plasma membrane. It is possible the Tcb proteins function in multiple steps monitored by the cellular PS transfer assay. The Tcb proteins interact with Sfk1 as well as flippases and floppases at the PM (see Manford et al. 2012 and SGD for additional citations). Thus, the Tcbs may have roles in transbilayer lipid organisation at the PM and the transfer of lyso-PS from the PM to the ER (given the published role of SMP proteins in recycling diacylglycerol to the ER) as well as anterograde transport of PS from the ER to the PM. We think that potential roles of the Tcbs in each of these transport steps between the PM and ER should be rigorously and carefully examined, but are beyond the scope of the current study.

While our results implicate the Tcb proteins in PS distribution, they do not rule out other functions for the Tcbs and we do not state that transfer of PS from the ER to the PM is their sole function.

If the Tcbs do not mediate PS transport, some mechanistic insight into how they affect PS distribution would significantly increase the impact of this study.

Response: Numerous studies have already shown that SMP domains directly transport phospholipids *in vitro* (Schauder et al., 2014; Saheki et al., 2016; Yu et al., 2016; Bian et al., 2018; Bian and De Camilli, 2019; Qian et al., 2021). Notably, this includes a recent study demonstrating that the Tcb3 SMP domain directly transports phospholipids *in vitro* (Qian et al., 2021). Our study now shows that the Tcb proteins

regulate PS distribution *in vivo*. PS clearly accumulates at the ER upon loss of the Tcb proteins. Moreover, we show that deletion of the Tcb3 SMP domain results in PS distribution defects upon heat stress.

The Osh6 and Osh7 proteins are established as PS transfer proteins based on similar criteria. They directly transfer PS *in vitro* and regulate PS distribution *in vivo*.

It is reasonable to speculate that the Tcb proteins directly transfer PS *in vivo*. We also agree with the reviewer that it would be interesting to explore additional mechanisms by which Tcbs might regulate PS metabolism and distribution, as discussed above. Our study draws functional links between the Tcb proteins and the Sfk1 protein that has been implicated in the control of transbilayer lipid organisation at the PM. Tcb3 may also interact with PM flippases and even be involved in transfer of lyso-PS from the PM to the ER. Exploring potential Tcb functions in these processes will be interesting and important for future studies; however, we believe this is outside of the scope of the current study.

Similarly, the study now makes a stronger case that the Tcbs are functionally linked to Pkh1 and Skh1, but there is still no mechanistic insight into how this occurs.

Response: Our revised manuscript does not include data regarding Pkh1 localisation. Instead, we demonstrate that the Tcb proteins regulate the *bona fide* PS effector protein Pkc1 that is required for PM integrity (see Roelants et al., *Biomolecules*, 2017 and additional references cited in our revised manuscript). We find that the Tcb proteins are required for efficient recruitment of Pkc1 to the mother cell cortex upon heat stress. Moreover, we show that overexpression of Pkc1 partially rescues the heat-induced PM integrity defects in cells lacking the Tcb proteins, demonstrating that Pkc1 function is compromised upon loss of the Tcb proteins. These results provide new (and mechanistic) understanding into the roles of the Tcb proteins in PM integrity during membrane stress conditions.

We also provide some mechanistic insight into the link between Sfk1 and Tcb3. We show that (1) the cytoplasmic C-terminal domain of Sfk1 is required for co-localisation with Tcb3 and (2) that deletion of the Sfk1 cytoplasmic C-terminal domain results in decreased localisation of the PS probe at the PM upon heat stress, similar to cells lacking Tcb3. In addition, both Sfk1 and the Tcb proteins are required for efficient recruitment of Pkc1 to the mother cell cortex upon heat stress. We also show that the Tcb proteins regulate sterol homeostasis at the PM, as has been recently reported for Sfk1 (Kishimoto et al., 2021). Thus, we not only describe a novel association between Sfk1 and the Tcb proteins under stress conditions but also draw links between their functions in Pkc1 localisation and PM lipid organisation.

In summary, several lines of convincing evidence suggest that the Tcb proteins transport PS and regulate PS distribution. These include:

- 1) the Tcb3 SMP domain directly transfers phospholipids *in vitro* (Quon et al., 2021)
- 2) PM-specific, but not ER-specific, PS species are significantly depleted in Δ tether cells (this study)

3) localisation of a PS reporter is decreased at the PM and increased at the ER in *tcb3* mutant cells (this study)

4) localisation and function of the PS-binding protein Pkc1 is impaired upon loss of the Tcb proteins (this study).

Reviewer #2 (Comments to the Authors (Required)):

The authors have added a considerable amount of additional evidence to show the role of Tricalbins and their role in PS transport. The work is carefully done with a number of controls. The work brings fresh insight into the role of Tricalbins.

Response: We are grateful for the positive response from Reviewer #2 who finds that our revised manuscript presents a “considerable amount of additional evidence” that “brings fresh insight” into the roles of the Tcb proteins.

April 4, 2022

RE: Life Science Alliance Manuscript #LSA-2022-01430R

Dr. Christopher J Stefan
University College London
Laboratory for Molecular Cell Biology
Gower Street
London, Greater London WC1E 6BT
United Kingdom

Dear Dr. Stefan,

Thank you for submitting your Research Article entitled "Tricalbin proteins regulate plasma membrane phospholipid homeostasis". It is a pleasure to let you know that your manuscript is now accepted for publication in Life Science Alliance. Congratulations on this interesting work.

DISTRIBUTION OF MATERIALS:

Again, congratulations on a very nice paper. I hope you found the review process to be constructive and are pleased with how the manuscript was handled editorially. We look forward to future exciting submissions from your lab.

Sincerely,
